# Towards A Generalist Code Embedding Model Based On Massive Data Synthesis

**Chaofan Li**[1,2*] **Jianlyu Chen**[1,3*]  **Yingxia Shao**[2,5†] **Defu Lian**[3†]  **Zheng Liu**[1,4†]

[1] Beijing Academy of Artificial Intelligence
[2] Beijing University of Posts and Telecommunications
[3] University of Science and Technoly of China
[4] The Hong Kong Polytechnic University
[5] Inspur Computer Technology Co., Ltd

{cfli, shaoyx}@bupt.edu.cn   chenjianlv@mail.ustc.edu.cn
liandefu@ustc.edu.cn   zhengliu1026@gmail.com

## Abstract

Code embedding models attract increasing attention due to the widespread popularity of retrieval-augmented generation (RAG) in software development. These models are expected to capture the rich semantic relationships inherent to code, which differ significantly from those found in text. However, existing models remain severely limited due to the scarcity of high-quality training data. In this work, we introduce **CodeR** (Code Retrieval), a state-of-the-art embedding model for general-purpose code retrieval. The superior performance of CodeR is built upon **CodeR-Pile**, a large-scale synthetic dataset constructed under the DRU (Diversity, Reliability, Usability) principle via a novel data synthesis pipeline. To optimize training effectiveness, we propose **Annealing**, a curriculum learning strategy that enables effective knowledge transfer across heterogeneous sources of data. We evaluate CodeR based on 16 diverse code retrieval tasks, where it significantly outperforms existing baselines and exhibits strong out-of-domain generalization performance. We have publicly released our code and the well-trained model to facilitate further research in this critical area[3].

## 1   Introduction

Thanks to the rapid advancement of large language models (LLMs), retrieval-augmented generation (RAG) has emerged as a promising paradigm for code development tasks, such as code completion and bug fixing [1]. Unlike traditional generation methods that make direct responses to the input prompts [2, 3], RAG enhances LLMs' performance by incorporating relevant context retrieved from internal code-bases or external knowledge sources. This additional context enables LLMs to generate more accurate and reliable code upon user's request. A common retrieval strategy involves vectorizing knowledge sources using a embedding model and retrieving relevant information based on embedding similarity [4, 5, 6, 7]. To achieve precise retrieval result, it is crucial to develop generalist code embedding models capable of capturing a wide range of semantic relationships [8, 9]. While significant progress has been made in the text field [10], high-quality code embedding models remain scarce due to a lack of suitable training data. In particular, most existing training datasets are tailored for text-centric scenarios like web search [11] and question answering [12]. These scenarios

---

[*]Equal contribution

[†]Corresponding authors, with Zheng Liu as the project lead.

[3]https://github.com/FlagOpen/FlagEmbedding/tree/master/research/BGE_Coder

Table 1: Comparison between CodeR-Pile and two existing datasets: CodeSearchNet (CSN) [13] and CoIR [14]. **#T**: number of code retrieval tasks. **#PL**: number of programming languages. **Size**: number of training samples. CoIR (new) refers to the newly introduced data samples in CoIR excluding those already introduced by CodeSearchNet.

| Dataset | Task Type | #T | Lan. | #PL | Size |
|---------|-----------|-----|------|-----|------|
| CSN | Text2Code | 1 | EN | 6 | 1.9M |
| CoIR (new) | Text2Code, Code2Code, Hybrid | 8 | EN | 11 | 307K |
| **CodeR-Pile** | Text2Code, Code2Text, Code2Code, Hybrid | 47 | EN, CN | 20 | 2.9M |

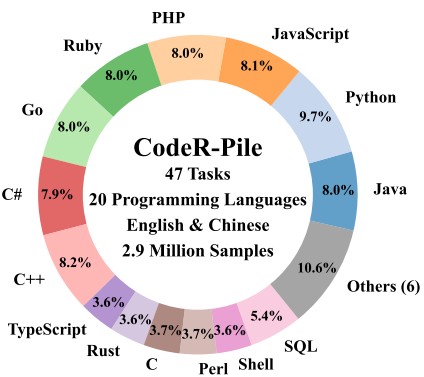

Figure 1: Programming languages distribution of CodeR-Pile.

are fundamentally different from the code-related ones in terms data forms and nature of semantic relationships, making them insufficient for code retrieval tasks.

In this paper, we introduce **CodeR**, a state-of-the-art code embedding model trained on a large-scale synthetic dataset named **CodeR-Pile**. To enhance the quality of training data, we propose a novel data synthesis strategy guided by the DRU principle, which emphasizes: 1) Diversity, ensuring comprehensively coverage of a wide range of data formats and semantic relationships; 2) Reliability, maintaining the correctness of semantic relationships; and 3) Usability, providing rich-semantic training samples for embedding models. With this principle, we design a novel data synthesis workflow which starts with high-level planning of task types and demonstrations, followed by the generation of training samples for each planned task. This progressive approach enables the creation of high-quality data that captures a wide variety of semantic relationships. Furthermore, we combine the use of expensive proprietary LLMs for high-level planning with lightweight open-source LLMs for training sample generation, which substantially improves the cost-effectiveness of the overall data synthesis process. As shown in Table 1 and Figure 1, CodeR-Pile exhibits three key advantages over existing datasets: 1) **task diversity**, with 4 major task types and 47 code retrieval tasks included; 2) **language coverage**, covering 20 popular programming languages (detailed in Appendix D) and 2 common natural languages (English, Chinese); and 3) **scale**, with 2.9 million training samples generated. Note that CodeR-Pile can be readily extended for higher diversity and scale with our open-sourced data synthesis workflow.

We integrate both text training data (which is abundant and readily available) and existing code-retrieval data (limited but useful) for the complement of the synthetic dataset. To make effective use of this heterogeneous data, we propose a curriculum learning strategy, called **Annealing**. This strategy follows a weak-to-strong supervision paradigm, where the training process begins with data of lower relevance and progressively shifts toward data of higher relevance. Specifically, the model is first trained on text-only data, characterized by low relevance and low entropy. This is followed by a mixture of text and code data, which introduces higher entropy and improved relevance. Finally, the model is fine-tuned on code-only data, which offers the highest relevance while maintaining low entropy. This approach enables smooth and effective knowledge transfer across different data domains while preserving the stability of training process.

For the sake of rigorous evaluation, we perform comprehensive experiments based on 16 diverse datasets from two popular code-retrieval benchmarks, CoIR [14] and CodeRAG [1]. We primarily use the CoIR for in-domain evaluation, which covers 4 main categories of code-retrieval scenarios and 14 programming languages. Meanwhile, we leave CodeRAG for out-of-domain evaluation, which contains challenging and distinct code-retrieval tasks from CoIR. Notably, the CodeR significantly outperforms existing code-focused and general-purposes embedding models. Meanwhile, the empirical advantages becomes more pronounced for those out-of-domain and complex scenarios, which further highlights the effectiveness of CodeR.

To summarize, the following contributions are made in this paper: 1) we introduce CodeR, a state-of-the-art code embedding model, and CodeR-Pile, a large-scale synthetic dataset that serves as its foundation; 2) we propose a novel curriculum learning strategy which enables effective utilization of heterogeneous training data; 3) we perform comprehensive evaluations which fully demonstrate the effectiveness of CodeR and the value of CodeR-Pile. The entire dataset and its curation pipeline will be publicly released along with the well-trained embedding model, which will provide valuable resources for the future progress of code retrieval and CodeRAG research.

## 2    Related Work

• **Data Synthesis for IR**. Synthetic data generation has become an increasingly popular practice in the field of information retrieval (IR) [15, 16, 17, 18]. In recent years, synthetic data has played a crucial role in training advanced, general-purpose retrieval models that demonstrate strong performance across a wide range of tasks. For example, Doc2query [19] and Promptagator [20] utilize language models to generate synthetic queries for unlabeled documents, thereby enhancing retrieval effectiveness in tasks, like document retrieval and question answering. E5-Mistral [21] introduces a "brainstorm" strategy, where the model autonomously generates tasks and task-relevant queries, along with positive and negative examples, thereby enabling more efficient and robust training. Similarly, Gecko [22] adopts a distillation-based approach for synthetic data generation, transferring knowledge from large, versatile large language models to more compact text embedding models.

However, existing synthetic IR datasets are primary focused on textual domains and fall short in code retrieval tasks. Moreover, current data synthesis methods are specifically designed for text-based retrieval scenarios, which typically consider a narrow scope of task types such as document retrieval and question answering. In contrast, code retrieval involves a diverse set of semantic matching tasks that are unique to programming languages and software development. These distinctions highlight the need for specialized data synthesis methods tailored to the requirements of code retrieval.

• **Code Retrieval**. Embedding models have made significant progress, driven by larger backbone encoders and the expansion of training scale [4, 5, 6, 7]. Recently, these models have demonstrated impressive performance on text-based benchmarks such as BEIR [10] and MTEB [23]. However, they still struggle with code retrieval tasks, as highlighted by recent empirical studies [14, 1]. While code generation has advanced through methods like CodeRL [24], which uses reinforcement learning with pre-trained models to enhance the generation of executable code, and CodeT [25], which generates multiple code samples and selects the best solution using test cases to boost performance, research on code retrieval is less mature. This gap is primarily due to a mismatch between the semantic structures of code and natural language, as well as the limited availability of training data specifically tailored for code retrieval. To alleviate these problems, several tailored code embedding models have been developed by different research teams. Among them, one of the most competitive is Voyage-v3 [26], which demonstrates substantially improved performance on public benchmarks like CoIR [14] and CodeRAG [1]. Unfortunately, both the model and its training data remain proprietary, limiting the community to build upon its progress. Meanwhile, other open-source models like CodeBERT [27] and GraphCodeBERT [28], as well as UniXcoder [29], Jina-v2-code [8], CodeSage-large-v2 [30] and CodeXEmbed [9] exhibit sub-optimal performance compared to Voyage-v3 and even fall behind some text-oriented approaches. Moreover, their training data is also not publicly released. Thus, the creation and open release of high-quality training data has become imperative for advancing code retrieval research.

## 3    Data Synthesis

Recent studies highlight the value of synthetic data for embedding models [31, 6, 5, 18]. Despite using various strategies, these works share several high-level principles regarding the effectiveness of synthetic data: (1) being Diverse and comprehensive, (2) introducing Reliable semantic relationships, (3) providing Useful and non-trivial knowledge. However, these works also reveal open problems in existing practice, e.g., directly prompting LLMs to generate training samples often results in a lack of diversity, while smaller LLMs may struggle with instruction-following capabilities.

In our work, we create a large-scale dataset, **CodeR-Pile**, through a novel data synthesis pipeline, adhering to the DRU principle and effectively collaborating large-small LLMs. As shown in Figure 2, the data synthesis pipeline includes three stages: 1) Brainstorming: designing diverse code retrieval

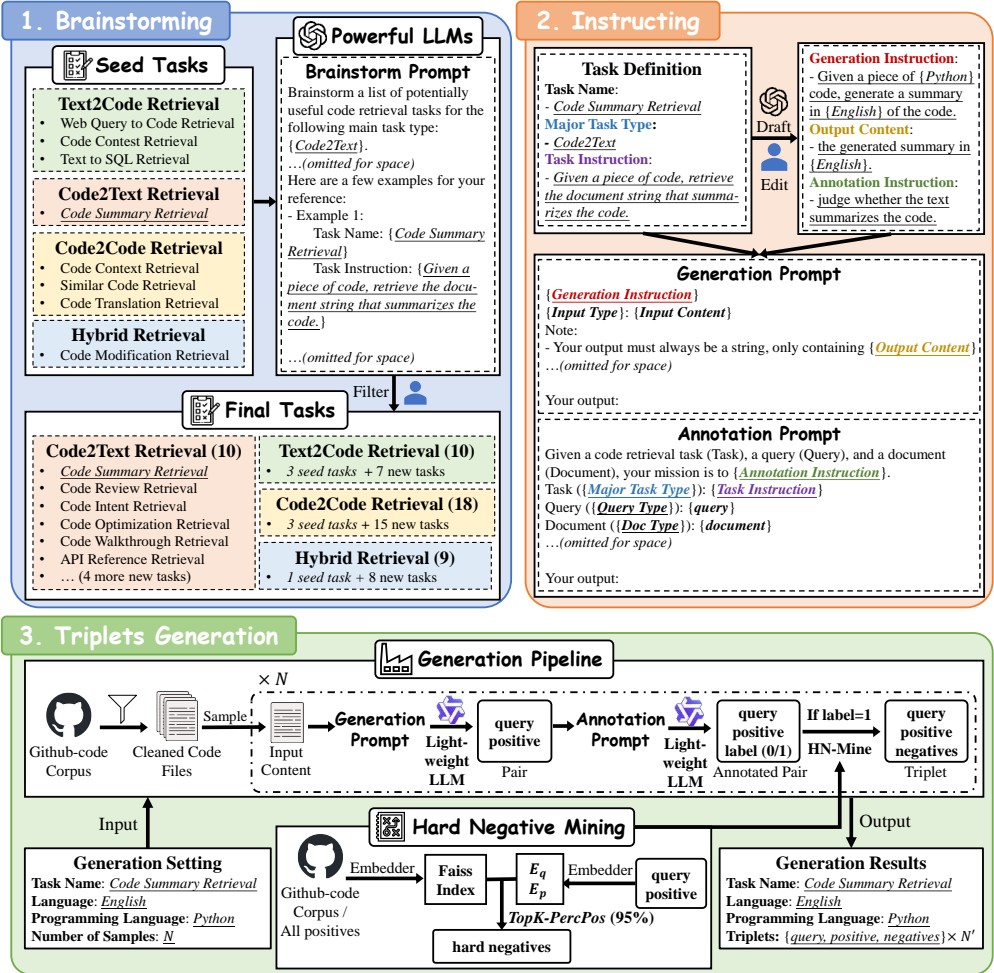

Figure 2: The data synthesis pipeline of **CodeR-Pile**.

tasks with *large and powerful LLMs*; 2) Instructing: defining generation and annotation instructions for each task with *large and powerful LLMs*; and 3) Triplets Generation: Generating training sample for each task with *cost-effective LLMs*. The detailed process is elaborated as follows.

**Brainstorming**. Instead of directly prompting LLMs to generate training samples, we first leverage the creativeness from large-and-powerful LLMs to design diverse training tasks. In our paper, the following four main categories are defined: 1) Text2Code; 2) Code2Text; 3) Code2Code; 4) Hybrid. We also manually design exemplars as the demonstrates to facilitate the brainstorming process. We make use of 4 powerful LLMs, including DeepSeek-R1 [32], GPT-4o [33], Claude-3-Opus-20240229 [34], and Gemini-2.0-Flash-Thinking-Exp [35]. For each task type, we manually check the brainstorming results, and filter out unqualified and duplicated results. Finally, this results in 10 tasks for Text2Code, 10 for Code2Text, 18 for Code2Code, and 9 for Hybrid. The detailed task types and task instructions are presented in Appendix D.

**Instructing**. Given the complexity of each designed task, it's still challenging to generate qualified training samples merely from the task's definition, especially with lightweight LLMs. Thus, we further use GPT-4o mini to draft detailed instructions, including task-specific generation instructions, output content requirements, and label-annotating instructions, to facilitate generation (Figure 2). The instructions are also refined by human experts to guarantee the quality. The detailed crafted instructions for all tasks are available in Appendix E.

**Triplets Generation**. We further design the generation pipeline to produce training samples (triplets) based on the well-defined task types and instructions. The generation process is iteratively conducted

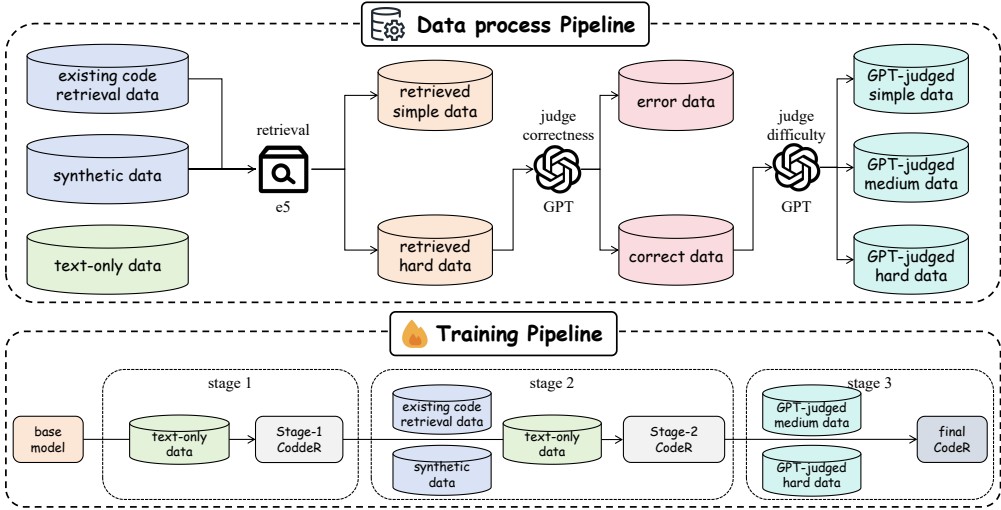

Figure 3: The training pipeline of CodeR.

through the following process. 1) Sample a code file from Github-code dataset[4] w.r.t. the specified programming language (e.g., Python). 2) Format the generation prompt based on the code-file and crafted instructions of the specified task. 3) Prompt a lightweight LLM, Qwen2.5-Coder-32B-Instruct [36] in our work, to generate the required output with the formatted prompt. The code-file and the LLM's output forms a **query-positive** pair for the corresponding task. 4) Format the **annotation prompt** with the organized query-positive pair and the crafted instructions of the specified task. 5) Quality control using Qwen2.5-Coder-32B-Instruct, which annotates each query-positive pair with 0/1 relevance label (1: true positive, 0: false positive). 6) Introducing 15 hard negatives for each verified positive sample [37].

Finally, CodeR-Pile dataset covers 4 major task types, 47 code retrieval tasks, 2 natural languages and 20 programming languages, including a total of 2,885,059 training samples. We provide concrete examples in Appendix D and more detailed specifications in Appendix E.

# 4 Training Method

Our code training process incorporates heterogeneous sources: text-only data and code data. The code data consists of existing code retrieval data and synthetic data. To make effective use of heterogeneous data and fully enhance the model's code retrieval capabilities, we propose a three-stage curriculum learning strategy [38] called **Annealing**. As shown in Figure 3, our training procedure follows a weak-to-strong supervision paradigm, beginning with low-relevance data and gradually transitioning to high-relevance data until the model training is complete. This curriculum learning strategy facilitates the transfer of knowledge from textual domains to programming languages, while progressively enhancing the model's ability to develop robust code retrieval capabilities.

**Stage 1: Warming-up with text data.** In this first stage, the base model is trained exclusively on text-only data. This stage has the most remote relevance to code retrieval, whose main purpose is to equip the model with fundamental semantic matching capabilities.

**Stage 2: Intensive training with all data.** In the second stage, the training data expands substantially to incorporate a mixture of text and code data. This introduces the highest diversity of data and improved relevance with code retrieval tasks. This modal-mixture training approach enables the model to transfer smoothly from text-only to text-and-code scenarios.

**Stage 3: Cooling-down with code data.** In the third stage, the training process focuses exclusively on code-only data, thereby strengthening the model's code retrieval capabilities. To enhance the training effectiveness, we implement the following strategies to emphasize more challenging tasks.

---

[4]https://huggingface.co/datasets/codeparrot/github-code-clean

Table 2: Results on the CoIR benchmark (NDCG@10).

| Model | Apps | CosQA | Text2SQL | CSN | CSN-CCR | CodeTrans -Contest | CodeTrans -DL | StackOverFlow QA | CodeFeedBack -ST | CodeFeedBack -MT | Avg |
|---|---|---|---|---|---|---|---|---|---|---|---|
| *Baselines* | | | | | | | | | | | |
| BGE-M3 | 7.37 | 22.73 | 48.76 | 43.23 | 47.55 | 47.86 | 31.16 | 51.04 | 49.94 | 33.46 | 39.31 |
| E5-Mistral-7B-Instruct | 21.33 | 31.27 | 65.98 | 54.25 | 65.27 | 82.55 | 33.24 | 91.54 | 72.71 | 33.65 | 55.18 |
| Gemini-embedding | 93.75 | **50.24** | 69.96 | 81.06 | 84.69 | 89.53 | 31.47 | 96.71 | 85.33 | 56.28 | 73.90 |
| UniXcoder | 1.36 | 25.14 | 50.45 | 60.20 | 58.36 | 41.82 | 31.03 | 44.67 | 36.02 | 24.21 | 37.33 |
| Jina-v2-code | 16.32 | 40.95 | 44.18 | 83.95 | 82.72 | 86.61 | 30.46 | 89.35 | 68.95 | 52.11 | 59.56 |
| CodeSage-large-v2 | 50.49 | 32.87 | 60.12 | 82.79 | 85.72 | 88.22 | 32.71 | 79.41 | 71.32 | 57.16 | 64.08 |
| CodeXEmbed-2B (general) | 74.99 | 36.31 | 59.00 | 73.50 | 85.77 | 86.63 | 33.17 | 90.54 | 81.15 | 53.08 | 67.41 |
| CodeXEmbed-7B (general) | 85.22 | 33.27 | 64.57 | 78.84 | 86.77 | 90.64 | 32.31 | 94.25 | 80.93 | 57.83 | 70.46 |
| CodeXEmbed-2B (in-domain) | 76.86 | 40.47 | 78.42 | 87.87 | 97.66 | 90.30 | 38.57 | 94.47 | 86.36 | 65.51 | 75.65 |
| CodeXEmbed-7B (in-domain) | 85.38 | 42.47 | **78.94** | **89.67** | 97.95 | 94.45 | 40.46 | 96.33 | 87.53 | 68.83 | 78.20 |
| Voyage-Code-002 | 26.52 | 29.79 | 69.26 | 81.79 | 73.45 | 72.77 | 27.28 | 67.68 | 65.35 | 28.74 | 56.26 |
| Voyage-Code-003 | 93.62 | 34.45 | 62.87 | 89.35 | 90.05 | **94.96** | 38.57 | **97.17** | 90.67 | 93.58 | 78.53 |
| *Ours* | | | | | | | | | | | |
| CodeR-1.5B + text-only data & full code data | **98.08** | 46.72 | 64.35 | 89.53 | **98.30** | 94.38 | 46.13 | 95.35 | 90.56 | **94.38** | **81.77** |
| w/ text-only data & synthetic data | 55.70 | 37.52 | 57.05 | 76.49 | 92.83 | 91.65 | 33.67 | 93.08 | 82.95 | 80.25 | 70.12 |
| w/ text-only data & existing code retrieval data | 81.45 | 46.56 | 64.20 | 89.06 | 97.84 | 94.42 | **46.27** | 95.12 | 89.63 | 93.20 | 79.77 |

1) We utilize E5 embedding model [39] (multilingual-e5-base) to perform retrieval for each query in the dataset. If the positive instance appears within the top-3 retrieved results, the sample is classified as simple and excluded from the dataset. Otherwise, it is considered as a hard sample and retained.

2) We leverage GPT-4o mini [33] for further refinement. Particularly, GPT-4o mini is prompted to labeled the data into four categories: "error data" (the annotation is considered problematic), "GPT-judged simple data", "GPT-judged medium data", and "GPT-judged hard data". We only retain the "GPT-judged medium data" and "GPT-judged hard data" for the last-stage's training. The detailed filter instructions are presented in Appendix F.

This rigorous selection process yields a training corpus consisting primarily of highly reliable and sufficiently challenging training samples. This helps to significantly enhance the model's capability to handle complex code retrieval scenarios. Additionally, we further add tasks instructions for the remaining training samples, which enables the model to establish stronger task-specific retrieval capability [40, 21]. The contrastive learning process, the instructions, and the detailed specifications are included in Appendix B. Besides, we provide details about training data and implementations in Appendix C and D.

## 5 Experiment

### 5.1 Benchmarks

Our experiment leverages the 16 diverse datasets from the two popular code-retrieval benchmarks: CoIR and CodeRAG. Note that CodeRAG has no overlap with the training data, thus offering out-of-domain evaluation for code embedding models.

**CoIR** [14] is a comprehensive benchmark designed to evaluate code retrieval performance. It consists of 10 datasets spanning 8 different retrieval tasks across 7 diverse domains. In our experiments, we evaluate on the test sets of each dataset included in the benchmark.

**CoIR-filter** is a refined version of the original CoIR. The original CoIR test sets contain a substantial number of redundant queries and duplicated corpus entries, which can undermine the reliability of evaluation. To mitigate this issue, we apply a de-duplication process to both queries and documents, yielding a cleaner and more trustworthy benchmark for assessing the performance.

**CodeRAG** [1] is a benchmark designed for evaluating retrieval-augmented code generation methods. The benchmark encompasses both retrieval and end-to-end code generation performance. We employ its retrieval session for experiment, which includes 6 code datasets.

### 5.2 Main Results

The experimental results on the CoIR is shown in Table 2. We conduct comprehensive comparisons against a wide range of baselines, categorized as follows: 1) General-purposes embedding models: BGE-M3 [41], E5-mistral-7B-instruct [21], and Gemini-embedding [7]. 2) Code-focused models: UniXcoder [29], Jina-v2-code [8], CodeSage-large-v2 [30], CodeXEmbed [9], and Voyage [26].

Table 3: Results on the CoIR-filter benchmark (NDCG@10).

| Model | Apps | CosQA | Text2SQL | CSN | CSN-CCR | CodeTrans | | StackOverFlow QA | CodeFeedBack | | Avg |
|---|---|---|---|---|---|---|---|---|---|---|---|
| | | | | | | -Contest | -DL | | -ST | -MT | |
| *Baselines* | | | | | | | | | | | |
| Jina-v2-code | 16.33 | 73.94 | 62.32 | 83.95 | 82.89 | 86.61 | 64.12 | 89.35 | 71.43 | 52.11 | 68.31 |
| CodeXEmbed-2B (general) | 80.68 | 64.13 | 84.25 | 73.53 | 87.69 | 83.31 | 70.11 | 91.50 | 81.79 | 49.09 | 76.61 |
| Voyage-code-003 | 93.72 | 64.00 | 94.78 | 89.35 | 94.17 | **94.96** | 71.96 | **97.17** | **93.66** | 93.58 | 88.73 |
| *Ours* | | | | | | | | | | | |
| CodeR-1.5B + text-only data & full code data | **98.18** | **78.02** | **95.87** | **89.53** | **98.51** | 94.33 | 92.30 | 95.35 | 93.53 | **94.38** | **93.00** |
| w/ text-only data & synthetic data | 55.75 | 63.58 | 83.42 | 76.49 | 93.02 | 91.65 | 71.83 | 93.08 | 80.25 | 85.91 | 79.50 |
| w/ text-only data & existing code retrieval data | 81.65 | 78.38 | 95.34 | 89.13 | 98.09 | 94.56 | 92.42 | 95.11 | 92.69 | 93.45 | 91.08 |

Table 4: Results on the CodeRAG benchmark (NDCG@10).

| Model | HummanEval | MBPP | DS-1000 | ODEX | RepoEval | SWE-bench-Lite | Avg |
|---|---|---|---|---|---|---|---|
| *Baselines* | | | | | | | |
| BGE-base | 99.7 | 98.0 | 10.8 | 22.0 | 77.5 | 44.9 | 58.5 |
| SFR-mistral-7B | 100.0 | 99.0 | 19.3 | **37.1** | 83.8 | 62.7 | 67.0 |
| Jina-v2-code | 100.0 | 97.7 | 26.2 | 19.9 | 90.5 | 58.3 | 65.4 |
| CodeSage-large-v2 | 100.0 | 96.3 | 16.2 | 14.1 | 84.7 | 39.9 | 58.5 |
| CodeXEmbed-2B (general) | 100.0 | 97.4 | 25.4 | 23.9 | 88.7 | 52.4 | 64.6 |
| Voyage-code-002 | 100.0 | 99.0 | 33.1 | 26.6 | **94.3** | 29.1 | 63.7 |
| *Ours* | | | | | | | |
| CodeR-1.5B + text-only data & full code data | 100.0 | 99.2 | **40.8** | 36.1 | 93.1 | **67.4** | **72.8** |
| w/ text-only data & synthetic data | 100.0 | 98.1 | 37.3 | 31.3 | 91.4 | 65.9 | 70.7 |
| w/ text-only data & existing code retrieval data | 100.0 | 98.9 | 37.2 | 32.5 | 92.2 | 64.6 | 70.9 |

Note that CodeXEmbed involves in two variants: one version is trained exclusively on general data (general), while the other one incorporates CoIR in-domain training data (in-domain).

**CoIR results.** The key observations are presented as follows. First of all, when CodeR is trained with the entire code data, it significantly outperforms existing general-purpose and code-focused models, achieving state-of-the-art performance. This result basically demonstrates the effectiveness of our training process and the value of our synthetic data. Additionally, when using the full code data, CodeR substantially outperforms the model that is trained solely on synthetic data or existing code retrieval data, which validates the efficacy of incorporating synthetic data to enhance model performance. Finally, when CodeR is trained exclusively on synthetic data, it outperforms the comparably sized CodeXEmbed-2B (general) by 2.7 points, and attains performance comparable to CodeXEmbed-7B (general), which further validate the impact of our synthetic data.

**CoIR-filter results.** We identify severe duplications in several datasets of CoIR, including Apps, CosQA, Text2SQL, CSN-CCR, CodeTrans-DL, and CodeFeedBack-ST. With the removal of all duplications, we further evaluate the performance of CodeR on CoIR-filter against Jina-v2-code, Codexembed-2B (general) and Voyage-code-003 (another strong baseline, CodeXEmbed, is not included as it hasn't released its powerful 7B model yet). The refined benchmark leads to more reliable evaluation, and it demonstrates more significant advantage of CodeR over the baselines (improvement over the state-of-the-art model Voyage-code-003 is increased from 3.2 to 4.3 points).

**CodeRAG results.** We further investigate the generalization capability of CodeR by evaluating its out-of-domain performance using CodeRAG, with experiment results presented in Table 4 (we only include the reported baselines in CodeRAG's evaluation due to constraints on model availability and evaluation expense). It can be seen that CodeR demonstrates exceptional performance on CodeRAG, achieving a state-of-the-art average score of 72.8. Additionally, CodeR demonstrates superior performance even when trained exclusively on synthetic data, substantially outperforming current open-source code retrievers such as Jina-v2-code and CodeXEmbed-2B (general). Furthermore, when trained exclusively on synthetic data, CodeR demonstrates performance comparable to training solely on existing code retrieval data, it confirms that synthetic data provides robustness equivalent to manually annotated data. Finally, when utilizing the full code data, CodeR outperforms models trained on either data type individually, which indicates that the integration of synthetic data bolsters the model's ability to handle a wider range of code retrieval tasks.

## 5.3 Analysis of Synthetic Data

We perform comprehensive pilot studies on synthetic data to explore the following research questions: 1) **RQ1**: How does task coverage influence the retrieval performance of models on downstream

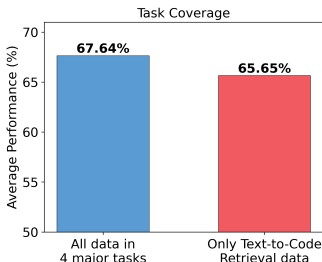 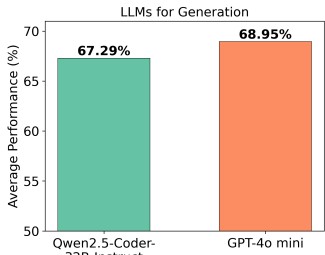 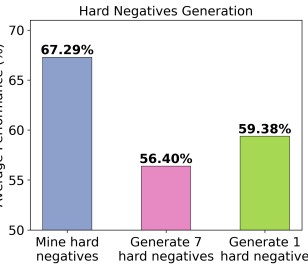

(a) Diverse training tasks enhance the model's code retrieval ability.

(b) Open-source lightweight LLMs are sufficient for generating quality training samples.

(c) Real-world corpus mining plays a critical role in enhancing the data synthesis pipeline.

Figure 4: Retrieval performance on the CoIR benchmark (NDCG@10) under different task coverage (Figure (a), RQ1), LLMs for generation (Figure (b), RQ2), and hard negatives strategy (Figure (c), RQ3), with detailed results provided in the Appendix H.

tasks? 2) **RQ2**: How do lightweight LLMs compare to powerful, proprietary LLMs in synthetic data generation? 3) **RQ3**: How do mined hard negatives compare to LLM-generated hard negatives in terms of data effectiveness? 4) **RQ4**: How reliable are relevance annotations generated by LLMs during the data synthesis process?

**Task Coverage (RQ1)**. To investigate how task coverage affect model's retrieval performance on downstream tasks, we directly fine-tune two models with different task coverage: 1) using all data in 4 major task types; 2) only using Text-to-Code Retrieval data. As shown in Figure 4 (a), the model using all data in 4 major task types exhibits better retrieval performance on CoIR benchmark (65.65 → 67.64), which indicates that more diverse task coverage is able to improve the model's retrieval capability in more diverse downstream tasks.

**LLMs for Generation (RQ2)**. To evaluate how lightweight LLMs compare to powerful yet expensive models in data generation, we select seven tasks[5] and use the same input content for both Qwen2.5-Coder-32B-Instruct and GPT-4o mini to generate training samples through our data synthesis pipeline. We then fine-tune two models using the training samples generated by each LLM, respectively. Figure 4 (b) shows the comparison results on the CoIR benchmark. Fine-tuning with GPT-4o mini's synthetic data yields only a 1.66 point improvement over using Qwen2.5-Coder-32B-Instruct's synthetic data. This suggests that lightweight open-source LLMs are sufficient for generating effective, high-quality training samples based on our proposed workflow.

**Hard Negatives Generation Strategy (RQ3)**. To assess the effectiveness of different hard negative generation strategies, we compare our approach that mines hard negatives from a real-world corpus, with the method proposed by E5-Mistral [31] which relies on LLM-generated hard negatives. Using the same seven tasks as in RQ2, we prompt Qwen2.5-Coder-32B-Instruct to generate $N$ hard negatives for each query-positive pair, replacing the mined hard negatives with those generated by the LLM. As shown in Figure 4 (c), fine-tuning with LLM-generated hard negatives results in significant drop in performance on CoIR benchmark, particularly for tasks such as Apps (Code Contest Retrieval) and CodeFeedBack-ST (Single-turn Code QA). These results highlight the crucial role of real-world corpus mining in our data synthesis pipeline.

**Reliability of Annotation (RQ4)**. To evaluate the annotation accuracy provided by LLMs during data generation, we first randomly sample 10 annotated query-positive pairs for each label (0 and 1) within a given task. These pairs are then reviewed by human experts with the assistance of powerful LLMs[6]. We conduct this evaluation using the generated data from the seven tasks in RQ2, and also annotate query-positive pairs generated by GPT-4o mini for comparison. As shown in Table 5, for pairs annotated as label = 1 (i.e., pairs included in the our dataset), Qwen2.5-Coder-32B-Instruct achieves an annotation accuracy of 93%, outperforming GPT-4o mini. This demonstrates the reliability of

---

[5]Text2Code: *Web Query to Code Retrieval*, *Code Contest Retrieval*, *Text to SQL Retrieval*. Code2Text: *Code Summary Retrieval*. Code2Code: *Code Context Retrieval*, *Similar Code Retrieval*, *Code Translation Retrieval*.

[6]We utilize DeepSeek-R1, GPT-4o, and Claude 3.7 Sonnet to assist human experts in annotating the provided query-positive pairs.

Table 5: Annotation accuracy of Qwen2.5-Coder-32B-Instruct and GPT-4o mini in data generation, with Qwen2.5-Coder-32B-Instruct demonstrating superior annotation accuracy for positive data pairs while maintaining stringent filtering criteria to ensure higher-quality datasets.

| Task (→) | Web Query to Code Retrieval | Code Contest Retrieval | Text to SQL Retrieval | Code Summary Retrieval | Similar Code Retrieval | Code Translation Retrieval | Avg |
|---|---|---|---|---|---|---|---|
| *Annotated Label = 1* | | | | | | | |
| Qwen2.5-Coder-32B-Instruct | 90% (9/10) | 100% (10/10) | 100% (10/10) | 100% (10/10) | 100% (10/10) | 70% (7/10) | 93% (56/60) |
| GPT-4o mini | 100% (10/10) | 80% (8/10) | 80% (8/10) | 100% (10/10) | 100% (10/10) | 70% (7/10) | 88% (53/60) |
| *Annotated Label = 0* | | | | | | | |
| Qwen2.5-Coder-32B-Instruct | 40% (4/10) | 60% (6/10) | 30% (3/10) | 10% (1/10) | 20% (2/10) | 70% (7/10) | 38% (23/60) |
| GPT-4o mini | 70% (7/10) | 80% (8/10) | 70% (7/10) | 33% (2/6) | 80% (8/10) | 90% (9/10) | 73% (41/56) |

Table 6: Abalation studies based on the CoIR benchmark (NDCG@10).

| Model | Apps | CosQA | Text2SQL | CSN | CSN-CCR | CodeTrans-Contest | CodeTrans-DL | StackOverFlowQA | CodeFeedBack-ST | CodeFeedBack-MT | Avg |
|---|---|---|---|---|---|---|---|---|---|---|---|
| *w/ text-only data & synthetic data* | | | | | | | | | | | |
| CodeR-1.5B | 55.70 | 37.52 | 57.05 | 76.49 | 92.83 | 91.65 | 33.67 | 93.08 | 82.95 | 80.25 | 70.12 |
| Stage-1 | 15.17 | 33.15 | 57.82 | 68.15 | 84.14 | 87.00 | 33.72 | 93.32 | 81.14 | 77.53 | 63.14 |
| Stage-2 | 53.28 | 36.35 | 54.87 | 75.24 | 92.43 | 90.81 | 33.66 | 92.58 | 79.96 | 82.07 | 69.13 |
| Stage-3 | 55.70 | 37.52 | 57.05 | 76.49 | 92.83 | 91.65 | 33.67 | 93.08 | 82.95 | 80.25 | 70.12 |
| w/o Annealing | 49.59 | 36.21 | 58.78 | 73.08 | 89.72 | 90.03 | 34.14 | 92.81 | 82.97 | 77.60 | 68.49 |
| w/o text-only data | 54.20 | 33.57 | 51.39 | 78.85 | 93.20 | 87.85 | 35.36 | 93.00 | 81.47 | 80.05 | 68.89 |
| w/o GPT filtering | 53.53 | 36.00 | 54.58 | 74.88 | 91.99 | 91.06 | 33.64 | 92.63 | 81.84 | 79.58 | 68.97 |
| w/o E5 and GPT filtering | 53.53 | 35.17 | 54.11 | 72.97 | 92.37 | 89.76 | 33.38 | 91.97 | 81.38 | 77.92 | 68.25 |

training samples generated by our data synthesis pipeline. However, for pairs annotated as label = 0 (i.e., pairs excluded from the final dataset), Qwen2.5-Coder-32B-Instruct's annotation accuracy drops to 38%, which is lower than GPT-4o mini. This indicates that the data filtering criteria under Qwen2.5-Coder-32B-Instruct are more stringent, further ensuring the quality of the data. Although these strict filtering criteria may result in some positive samples being filtered out, they ensure that negative samples are effectively discarded, guaranteeing that the retained data are truly high-quality and accurate.

## 5.4 Analysis of Training Process

**Performance Gain of Each Stage.** We evaluate the effectiveness of our training approach by analyzing the performance improvements of CodeR across each stage (Table 6). The results show consistent performance gains throughout the training process. Notably, the most significant improvement occurs during the second stage, which integrates all three data sources. This highlights the effectiveness of knowledge transfer from heterogeneous data to the code retrieval task.

**Effectiveness of Annealing.** We conduct a detailed analysis of Annealing using the following comparison methods: 1) w/o Annealing, which trains the model directly on mixed data from all three stages. 2) w/o text-only data, which removes text-only data and training with only code data in a two-stage process.

The experimental result is presented in Table 6. As observed, training with Annealing leads to significant improvements over direct mixed-data training, underscoring the importance of the three-stage training strategy. By following a weak-to-strong supervision paradigm, the model develops enhanced code retrieval capabilities, allowing it to better tackle complex code retrieval tasks.

Additionally, removing text-only data results in a notable performance drop, indicating that incorporating textual information is a critical component of code training. This textual data facilitates knowledge transfer that improves overall code retrieval performance.

**Impact of Filtering Strategy.** We investigate the impact of data filtering strategies in the third stage of Annealing by comparing our approach with the following methods: 1) w/o GPT filtering: using only the retrieved hard negatives for three-stage training. 2) w/o E5 and GPT filtering: using the original code data without any filtering for three-stage training.

The experimental results, as shown in Table 6, indicate that incorporating GPT filtering effectively improves the quality of training data, leading to enhanced retrieval performance. In contrast, training

directly on the unfiltered code dataset results in a significant decline in performance. This suggests that the original dataset contains low-quality data, and since the model has already acquired strong code retrieval capabilities, further training on the unfiltered data can hardly contribute to performance improvements.

## 6 Conclusion

In this paper, we present CodeR, a state-of-the-art embedding model designed for general-purpose code retrieval. CodeR's superior performance is built upon CodeR-Pile, a massive synthetic dataset generated through our innovative data synthesis workflow. To effectively leverage the heterogeneous training data, we propose a novel curriculum learning method called Annealing, which progressively trains the model using a weak-to-strong supervision strategy. Comprehensive experiments on CoIR and CodeRAG benchmarks demonstrate CodeR's significant advantage over existing models, highlighting the value of our training data and the effectiveness of our training approach.

## 7 Discussion on Limitations

While CodeR has made remarkable advancements in code retrieval, there remain several opportunities for further improvement. First, although CodeR currently supports code retrieval in both Chinese and English, expanding its capacity to cover additional languages would enhance its versatility. Additionally, while the existing 1.5B-scale model achieves impressive performance, developing models at varying scales could better accommodate diverse application scenarios. Finally, the synthetic data we have created holds potential to support the development of re-ranking models. We plan to explore these directions in our future research.

## 8 Acknowledgments

The work was supported by grants from the National Key R&D Program of China Grant No. 2023YFF0725103, the National Natural Science Foundation of China (No. U24A20253), the National Natural Science Foundation of China (Nos. 62272054, 62192784), Beijing Nova Program (No. 20230484319, 20250484968), State Key Laboratory of Multimedia Information Processing Open Fund (No. SKLMIP-KF-2025-07) and Shandong Key Laboratory of Advanced Computing.

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

# A  Overview of Appendix

# B  Instruction Tuning

Given a task definition $t$ and a query $q$, we construct an instructed query using the following formulation:

$$q_{inst} = \langle instruct \rangle \{t\} \langle query \rangle \{q\} \tag{1}$$

For encoding purposes, we append an [EOS] token to each instructed query $q_{inst}$ and document $d$. The respective embeddings $(h_{q_{inst}}, h_d)$ are extracted from the hidden states corresponding to the [EOS] token in the final layer of the model.

During training, we use the standard InfoNCE [42] contrastive loss function:

$$\mathcal{L} = -\log \frac{\exp(\text{sim}(q_{\text{inst}}, d^+))}{\exp(\text{sim}(q_{\text{inst}}, d^+)) + \sum_{d^- \in D^-} \exp(\text{sim}(q_{\text{inst}}, d^-))} \tag{2}$$

where $d^+$ represents a relevant document, and $D^-$ denotes the collection of irrelevant documents that serve as negative examples. The similarity function $\text{sim}(q_{inst}, d)$ quantifies the degree of matching between the instructed query $q_{inst}$ and the document $d$. Here we employ a temperature-scaled cosine similarity function defined as:

$$\text{sim}(q_{\text{inst}}, d) = \frac{1}{\tau} \cos(h_{q_{inst}}, h_d) \tag{3}$$

In this formulation, $\tau$ is a temperature hyperparameter that controls the sharpness of the distribution, which we fix at 0.02 throughout the training process.

# C  Experiment Setup

## C.1  Training Dataset

We utilize two distinct datasets for training: text-only data and code data. The code data is structured into two subsets: existing code retrieval data and synthetic data.

- For the text-only data, we incorporate retrieval datasets and STS datasets from the BGE training set [5, 41].
- For the existing code retrieval data, we adopt the CoIR training datasets from their original source [14].

## C.2  Implementation Details

CodeR is initialized Qwen-2.5-Coder-1.5B [36] as its base model. Throughout the training procedure, we set the maximum sequence length of 512 tokens for both queries and passages. Following previous works [31, 43], we utilize Low-Rank Adaptation (LoRA) [44] with a rank of 32 and an alpha of 64. During the first and second training stages, we employ a learning rate of $1 \times 10^{-4}$. For the third stage, we decrease the learning rate to $1 \times 10^{-5}$ to facilitate finer gradient updates and enhance convergence precision. The model is trained for 5 days with 8 × A800 GPUs.

## C.3  Artifacts

The model and dataset release information is available at `https://github.com/FlagOpen/FlagEmbedding`.

Table 7: Task names and task instructions for all 47 code retrieval tasks in CodeR-Pile. In the task instruction of Code Translation Retrieval, "*{src_code_language}*" and "*{tgt_code_language}*" refer to the source programming language and the target programming language, respectively.

| Task Name | Task Instruction |
| --- | --- |
| *Text2Code Retrieval (10)* | |
| Web Query to Code Retrieval | Given a web search query, retrieve relevant code that can help answer the query. |
| Code Contest Retrieval | Given a code contest problem description, retrieve relevant code that can help solve the problem. |
| Text to SQL Retrieval | Given a question in text, retrieve SQL queries that are appropriate responses to the question. |
| Error Message to Code Retrieval | Given an error message encountered during coding, retrieve relevant code that can help resolve the error. |
| Code Explanation to Implementation Retrieval | Given a textual explanation of code functionality, retrieve the corresponding code implementation. |
| API Usage Description to Code Retrieval | Given a usage description of an API or library, retrieve code examples demonstrating the usage. |
| Bug Description to Code Retrieval | Given a description of a software bug or unexpected behavior, retrieve relevant code that can help address the issue. |
| Pseudocode to Code Retrieval | Given a pseudocode description of a procedure, retrieve code implementations of the procedure. |
| Programming Tutorial Query to Code Example Retrieval | Given a query related to a programming tutorial or learning material, retrieve code examples that are relevant to the query. |
| Algorithm Description to Code Retrieval | Given a textual description of an algorithm, retrieve code implementations of the described algorithm. |
| *Code2Text Retrieval (10)* | |
| Code Summary Retrieval | Given a piece of code, retrieve the document string that summarizes the code. |
| Code Review Retrieval | Given a piece of code, retrieve the review that explains its role. |
| Code Intent Retrieval | Given a piece of code, retrieve the developer's intent or purpose described in a commit message or design document. |
| Code Optimization Retrieval | Given a piece of code, retrieve optimization suggestions or performance analysis reports. |
| Tutorial Retrieval | Given a piece of code, retrieve tutorials or how-to guides that demonstrate how to use or implement similar code. |
| Code Issue Discussion Retrieval | Given a piece of code, retrieve discussions or issue reports related to the code, such as bug reports or feature requests. |
| API Reference Retrieval | Given a piece of code that uses specific APIs or libraries, retrieve the relevant API reference documentation for those APIs or libraries. |
| Code Walkthrough Retrieval | Given a piece of code, retrieve a step-by-step walkthrough or detailed explanation of the code's logic and execution flow. |
| Code Error Explanation Retrieval | Given a piece of code, retrieve the document that explains potential errors or exceptions that may arise from the code. |
| Code to Requirement Retrieval | Given a piece of code, retrieve the software requirement or user story it fulfills. |
| *Code2Code Retrieval (18)* | |
| Code Context Retrieval | Given a piece of code segment, retrieve the code segment that is the latter part of the code. |
| Similar Code Retrieval | Given a piece of code, retrieve code that is semantically equivalent to the input code. |
| Code Translation Retrieval | Given a piece of *{src_code_language}* code, retrieve *{tgt_code_language}* code that is semantically equivalent to the input code. |
| Code Refinement Retrieval | Given a piece of code, retrieve a refined version of the code. |
| Secure Code Retrieval | Given a piece of code, retrieve a version of the code with enhanced security measures or vulnerability fixes. |
| Code Version Update Retrieval | Given a piece of code in an older language version, retrieve code updated to comply with the syntax or features of a newer language version. |
| Code Example Retrieval | Given a code library or API, retrieve example code snippets that demonstrate how to use the library or API. |
| Code Dependency Retrieval | Given a piece of code, retrieve all the code segments that the input code depends on, including libraries, functions, and variables. |
| Code Pattern Retrieval | Given a piece of code, retrieve other code segments that follow the same design pattern or structure. |
| Code History Retrieval | Given a piece of code, retrieve previous versions or iterations of the code to understand its development history |
| Code Integration Retrieval | Given a piece of code, retrieve code that demonstrates how to integrate the input code with other systems or components. |
| Optimized Code Retrieval | Given a piece of code, retrieve an optimized version of the code that improves performance, readability, or efficiency. |
| Code Simplification Retrieval | Given a complex piece of code, retrieve a simplified version of the code that is easier to understand and maintain. |
| Code Modularization Retrieval | Given a piece of code, retrieve a modularized version of the code that breaks it down into smaller, reusable components. |
| Code Augmentation Retrieval | Given a piece of code, retrieve code that implements additional functionality while preserving the original behavior. |
| Error Handling Retrieval | Given a piece of code, retrieve code that incorporates error-checking or exception-handling mechanisms relevant to the code. |
| Code Documentation Retrieval | Given a piece of code, retrieve code with inline comments or documentation explaining its functionality. |
| Library Adaptation Retrieval | Given a piece of code using one library or framework, retrieve code that achieves the same functionality using a different library or framework. |
| *Hybrid Retrieval (9)* | |
| Code Modification Retrieval | Given a code snippet and a natural language description of desired modifications, retrieve relevant code that implements the requested modifications. |
| Code Bug Fix Example Retrieval | Given a code snippet containing a bug and a natural language description of the bug or error, retrieve code snippets that demonstrate solutions or fixes for similar bugs or errors. |
| Code Refactoring Pattern Retrieval | Given a code snippet that could be improved and a natural language description of desired refactoring goals or patterns, retrieve code snippets that exemplify similar refactoring techniques or patterns. |
| Code Style Guideline Example Retrieval | Given a code snippet and a natural language query describing a desired coding style or best practice, retrieve code snippets that adhere to the specified style guidelines or best practices. |
| Code Migration Retrieval | Given a code snippet and a natural language description of a specific migration requirement, retrieve code snippets that demonstrate how to migrate the code to meet the requirement. |
| Code Optimization Hybrid Retrieval | Given a code snippet and a natural language request for specific optimization, retrieve relevant code that implements the requested optimization. |
| Code Comparison Retrieval | Given two code snippets and a natural language query about their differences or similarities, retrieve relevant documents that explain the differences or similarities. |
| Code Best Practices Retrieval | Given a code snippet and a natural language query about coding best practices, retrieve relevant documents including guidelines, design patterns, or recommendations that can help improve the quality of the code. |
| Security Vulnerability Fix Retrieval | Given a code snippet and a text description of a security concern, retrieve secure code alternatives that address the security vulnerability. |

# D  Details on CodeR-Pile Dataset

## D.1  Specifications

Table 7 presents task names and task instructions for all 47 code retrieval tasks in CodeR-Pile. We also provide descriptive statistics of CodeR-Pile in Table 8.

## D.2  Supported Programming Languages

CodeR-Pile consists of the following 20 programming languages: Java, Python, JavaScript, PHP, Ruby, Go, C#, C++, TypeScript, Rust, C, Perl, Shell, SQL, Visual Basic, Powershell, Batchfile, Fortran, Haskell, and Lua.

Table 8: Descriptive statistics of CodeR-Pile. **#Tasks**: number of tasks. **#Samples**: number of training samples. One sample consists of a query, a positive document and a list of negative documents. **Avg Q Len**: average number of tokens per query. **Avg D Len**: average number of tokens per document. GPT-4o's tokenizer is used to count tokens.

| Task Type | #Tasks | Language | #Samples | Avg Q Len | Avg D Len |
|---|---|---|---|---|---|
| Text2Code Retrieval | 10 | English | 353,801 | 89 | 212 |
| | | Chinese | 351,268 | 144 | 215 |
| Code2Text Retrieval | 10 | English | 457,032 | 250 | 200 |
| | | Chinese | 363,367 | 234 | 287 |
| Code2Code Retrieval | 18 | English | 741,294 | 227 | 226 |
| Hybrid Retrieval | 9 | English | 309,499 | 281 | 213 |
| | | Chinese | 308,798 | 286 | 223 |
| Total | 47 | - | 2,885,059 | 217 | 225 |

Table 9: Prompt template for brainstorming tasks. For the placeholders, *{Task Name of Example $i$}* and *{Task Instruction of Example $i$}* refer to the task name and task instruction of the $i$-th seed task, which are available in Table 7.

Brainstorm a list of potentially useful code retrieval tasks for the following major task type: *{Major Task Type}*.

Note:
- For each task, you should generate the task name and the corresponding task instruction.
- In the task instruction, you should specify what the query is, and what the desired documents are.
- Each task should cover a wide range of queries, and should not be too specific.

Here are a few examples for your reference:
- Example 1:
    Task Name: *{Task Name of Example 1}*
    Task Instruction: *{Task Instruction of Example 1}*

- Example 2:
    Task Name: *{Task Name of Example 2}*
    Task Instruction: *{Task Instruction of Example 2}*

(*More examples · · ·*)

Your output should always be a list of JSON objects only, with about 10 elements. Each element should be a JSON object, containing the following fields:
- "task_name": the name of the task.
- "task_instruction": the instruction of the task.

Your output should start with "[" and end with "]". Remember do not explain your output or output anything else.

Your output:

# E  Details on Data Synthesis Pipeline

**Brainstorming Tasks**. The prompt template for brainstorming tasks is presented in Table 9. We select 7 tasks from the three major task types introduced by CoIR [14] as the seed tasks: 1) Text2Code: Web Query to Code Retrieval, Code Contest Retrieval, Text to SQL Retrieval; 2) Code2Text: Code Summary Retrieval; 3) Code2Code: Code Context Retrieval. We found that the two original hybrid retrieval tasks in CoIR, Single-turn Code QA and Multi-turn Code QA, are too broad in scope. Therefore, we design a new hybrid retrieval task, Code Modification Retrieval, and use it as the seed task to help LLMs brainstorm additional tasks.

**Query-Positive Pair Generation**. The prompt template for generating query-positive pairs is presented in Table 10. After generation, the input content and the LLM's output are organized into a query-positive pair based on the task requirements. For example, if the task is Code Summary Retrieval, then the input content is treated as the **query**, and the LLM's output serves as the **positive**. If the task is *Text to SQL Retrieval*, then the input content (*SQL code*) serves as the **positive**, and the LLM's output (*text query*) corresponds to the **query**.

Table 10: Prompt template for generating query-positive pairs. "*{Generation Instruction}*" ∈ Table 12. "*{Output Content}*" ∈ Table 13.

---

*{Generation Instruction}*

*{Input Type}*:
```
*{Input Content}*
```

Note:
- Your output must always be a string, only containing *{Output Content}*.
- Your output should be independent of the given code, which means that it should not containing the pronouns such as "it", "this", "that", "the given", "the provided", etc.

Remember do not explain your output or output anything else.

Your output:

---

Table 11: Prompt template for annotating query-positive pairs. For placeholders, "*{Major Task Type}*" ∈ {Text2Code, Code2Text, Code2Code, Hybrid}, "*{Annotation Instruction}*" ∈ Table 14, "*{Task Instruction}*" ∈ Table 7, "*{Query Type} and {Doc Type}*" ∈ Table 15.

---

Given a code retrieval task (Task), a query (Query), and a document (Document), your mission is to *{Annotation Instruction}*.

Task (*{Major Task Type}*): *{Task Instruction}*

Query (*{Query Type}*):
```
*{query}*
```

Document (*{Doc Type}*):
```
*{document}*
```

Your output must be one of the following options:
- 0: The query or document does not match the major task type (*{Major Task Type}*).
- 1: The query and document match the major task type (*{Major Task Type}*). The judgment is: Yes.
- 2: The query and document match the major task type (*{Major Task Type}*). The judgment is: No.

Do not explain your answer in the output. Your output must be a single number (0 or 1 or 2).

Your output:

---

**Query-Positive Pair Annotation**. The prompt template annotation is presented in Table 11. If the response from LLM is "1", then the final label is also 1 (true positive). If the response is not "1", then the final label is 0 (false positive).

**Hard Negatives Mining**. We use Topk-PercPos [37] with a 95% margin to mine 15 hard negatives for each query-positive pair.

**Comparison with E5-Mistral's Approach**. Compared with the data synthesis approach proposed by E5-Mistral [31], our data synthesis method differs in three main aspects: 1) The synthesis of query-positive pair and the mining of hard negatives both exploit the real-world corpus, which not only closely aligns with real-world scenarios, but also significantly reduces the generation cost. 2) The design of human-checked tasks, human-edited instructions, and annotated query-positive pairs enhances the reliability of CodeR-Pile. 3) The combined use of powerful yet expensive LLMs and lightweight open-source LLMs substantially improves the cost-effectiveness of CodeR-Pile's overall synthesis process.

Table 12: Generation instructions for all 47 code retrieval tasks in CodeR-Pile, used in the generation prompt. "*{code_language}*", "*{src_code_language}*", "*{tgt_code_language}*" ∈ {Java, Python, JavaScript, PHP, Ruby, Go, C#, C++, TypeScript, Rust, C, Perl, Shell, SQL, Visual Basic, Powershell, Batchfile, Fortran, Haskell, Lua}. "*{language}*" ∈ {English, Chinese}.

| Task Name | Generation Instruction |
|---|---|
| *Text2Code Retrieval (10)* | |
| Web Query to Code Retrieval | Given a piece of *{code_language}* code, generate a web query in *{language}* that can be solved by the code. |
| Code Contest Retrieval | Given a piece of *{code_language}* code, generate a code contest description in *{language}* that can be solved by the code. |
| Text to SQL Retrieval | Given a piece of *{code_language}* code, generate a text query in *{language}* for which the code is the appropriate response. |
| Error Message to Code Retrieval | Given a piece of *{code_language}* code, generate a possible error message in *{language}* that can be resolved by the code. |
| Code Explanation to Implementation Retrieval | Given a piece of *{code_language}* code, generate a textual explanation in *{language}* of the code functionality. |
| API Usage Description to Code Retrieval | Given a piece of *{code_language}* code, generate a usage description of an API or library in *{language}* that can be demonstrated by the code as an example. |
| Bug Description to Code Retrieval | 1. Given a piece of *{code_language}* code, modify some details of the code to introduce one or more bugs. 2. Given a piece of *{code_language}* code with one or more bugs, generate a description of the bugs in *{language}*. |
| Pseudocode to Code Retrieval | Given a piece of *{code_language}* code, generate a pseudocode in *{language}* that describes the code functionality. |
| Programming Tutorial Query to Code Example Retrieval | Given a piece of *{code_language}* code, generate a programming tutorial query in *{language}* that can be answered by the code as an example. |
| Algorithm Description to Code Retrieval | Given a piece of *{code_language}* code, generate an algorithm description in *{language}* that can be implemented by the code. |
| *Code2Text Retrieval (10)* | |
| Code Summary Retrieval | Given a piece of *{code_language}* code, generate a summary in *{language}* of the code. |
| Code Review Retrieval | Given a piece of *{code_language}* code, generate a review in *{language}* that explains its role. |
| Code Intent Retrieval | Given a piece of *{code_language}* code, generate a developer's intent or purpose described in a commit message or design document in *{language}*. |
| Code Optimization Retrieval | Given a piece of *{code_language}* code, generate code optimization suggestions or performance analysis reports in *{language}*. |
| Tutorial Retrieval | Given a piece of *{code_language}* code, generate tutorials or how-to guides that demonstrate how to use or implement similar code in *{language}*. |
| Code Issue Discussion Retrieval | 1. Given a piece of *{code_language}* code, generate a version with some bugs. 2. Given a piece of *{code_language}* code, generate a discussion of the code's issues or bugs in *{language}*, such as bug reports or feature requests. |
| API Reference Retrieval | Given a piece of *{code_language}* code, generate the relevant API reference documentation in *{language}* that can be used to understand the code. |
| Code Walkthrough Retrieval | Given a piece of *{code_language}* code, generate a step-by-step walkthrough or detailed explanation of the code's logic and execution flow in *{language}*. |
| Code Error Explanation Retrieval | Given a piece of *{code_language}* code, generate a detailed explanation of the errors or exceptions that may arise from the code in *{language}*. |
| Code to Requirement Retrieval | Given a piece of *{code_language}* code, generate a software requirement or user story it fulfills in *{language}*. |
| *Code2Code Retrieval (18)* | |
| Code Context Retrieval | Given a piece of *{code_language}* code, generate a piece of code that is the latter part of the input code. |
| Similar Code Retrieval | Given a piece of *{code_language}* code, generate a piece of *{code_language}* code that is semantically equivalent to the input code. |
| Code Translation Retrieval | Given a piece of *{src_code_language}* code, generate a piece of *{tgt_code_language}* code that is semantically equivalent to the input code. |
| Code Refinement Retrieval | Given a piece of *{code_language}* code, generate a refined version of the code. |
| Secure Code Retrieval | Given a piece of *{code_language}* code, generate a a version of the code with enhanced security measures or vulnerability fixes. |
| Code Version Update Retrieval | 1. Given a piece of *{code_language}* code, generate a lower-level version of the code. 2. Given a piece of *{code_language}* code, update it with the syntax or features of a newer language version. |
| Code Example Retrieval | Given a piece of *{code_language}* code, generate a piece of *{code_language}* code that is a good example of the code's usage. |
| Code Dependency Retrieval | Given a piece of *{code_language}* code, generate the code segments that the input code depends on, including libraries, functions, and variables. |
| Code Pattern Retrieval | Given a piece of *{code_language}* code, generate a piece of *{code_language}* code that follows the same design pattern or structure. |
| Code History Retrieval | Given a piece of *{code_language}* code, generate a piece of *{code_language}* code that is a historical version or iteration of the code. |
| Code Integration Retrieval | Given a piece of *{code_language}* code, generate a piece of *{code_language}* code that integrates the input code with other systems or components. |
| Optimized Code Retrieval | Given a piece of *{code_language}* code, generate an optimized version of the code that improves performance, readability, or efficiency. |
| Code Simplification Retrieval | Given a piece of *{code_language}* code, generate a simplified version of the code that is easier to understand and maintain. |
| Code Modularization Retrieval | Given a piece of *{code_language}* code, generate a modularized version of the code that breaks it down into smaller, reusable components. |
| Code Augmentation Retrieval | Given a piece of *{code_language}* code, generate a piece of code that implements additional functionality while preserving the original behavior. |
| Error Handling Retrieval | Given a piece of *{code_language}* code, generate a piece of code that incorporates error-checking or exception-handling mechanisms relevant to the input code. |
| Code Documentation Retrieval | Given a piece of *{code_language}* code, generate a piece of code with inline comments or documentation explaining its functionality. |
| Library Adaptation Retrieval | Given a piece of *{code_language}* code, generate a piece of code that achieves the same functionality using a different library or framework. |
| *Hybrid Retrieval (9)* | |
| Code Modification Retrieval | 1. Given a piece of input code and a piece of output code, generate the differences in *{language}* between the input code and output code. 2. Given the differences in *{language}* between a piece of input code and a piece of output code, generate a code modification instruction in *{language}* that uses only the information from the differences to transform the input code into the output code. |
| Code Bug Fix Example Retrieval | 1. Given a piece of *{code_language}* code, generate a buggy version of the code and a description in *{language}* of the bug or error. 2. Given a piece of *{code_language}* code and a natural language description of the bug or error, generate a piece of *{code_language}* code that demonstrates a solution or fix for the bug or error. |
| Code Refactoring Pattern Retrieval | 1. Given a piece of *{code_language}* code, generate a description of the desired refactoring goals or patterns in *{language}*. 2. Given a piece of *{code_language}* code and a natural language description of the desired refactoring goals or patterns, generate a piece of *{code_language}* code that exemplifies similar refactoring techniques or patterns. |
| Code Style Guideline Example Retrieval | 1. Given a piece of *{code_language}* code, generate a query describing a desired coding style or best practice to improve it in *{language}*. 2. Given a piece of *{code_language}* code and a natural language query describing the desired style guidelines or best practices, generate a piece of *{code_language}* code that adheres to the specified style guidelines or best practices. |
| Code Migration Retrieval | 1. Given a piece of *{code_language}* code, generate a specific migration requirement in *{language}* based on the code. 2. Given a piece of *{code_language}* code and a natural language description of a specific migration requirement, generate a piece of *{code_language}* code that meets the migration requirement. |
| Code Optimization Hybrid Retrieval | 1. Given a piece of *{code_language}* code, generate a question in *{language}* that requests a specific optimization for the code. 2. Given a piece of *{code_language}* code and a natural language request in *{language}* for specific optimization, generate a piece of output code that implements the requested optimization. |
| Code Comparison Retrieval | 1. Given a piece of input code and a piece of output code, generate a question in *{language}* about their differences or similarities. 2. Given a piece of input code and a piece of output code, and a natural language question in *{language}* about their differences or similarities, generate a response that answer the question. |
| Code Best Practices Retrieval | 1. Given a piece of *{code_language}* code, generate a question in *{language}* about coding best practices related to the code. 2. Given a piece of *{code_language}* code and a natural language question in *{language}* about coding best practices related to the code, generate a response including guidelines, design patterns, or recommendations that can help improve the quality of the code. |
| Security Vulnerability Fix Retrieval | 1. Given a piece of *{code_language}* code, generate a text description in *{language}* of a possible security concern in the code. 2. Given a piece of *{code_language}* code and a text description in *{language}* of a security concern, generate secure code alternatives that address the vulnerability. |

Table 13: Output contents of generation for all 47 code retrieval tasks in CodeR-Pile, used in the generation prompt. For placeholders, "*{code_language}*", "*{src_code_language}*", "*{tgt_code_language}*" ∈ {Java, Python, JavaScript, PHP, Ruby, Go, C#, C++, TypeScript, Rust, C, Perl, Shell, SQL, Visual Basic, Powershell, Batchfile, Fortran, Haskell, Lua}, "*{language}*" ∈ {English, Chinese}.

| Task Name | Output Content |
|---|---|
| ***Text2Code Retrieval (10)*** | |
| Web Query to Code Retrieval | the generated web query in *{language}* |
| Code Contest Retrieval | the generated code contest description in *{language}* |
| Text to SQL Retrieval | the generated text query in *{language}* |
| Error Message to Code Retrieval | the generated error message in *{language}* |
| Code Explanation to Implementation Retrieval | the generated explanation in *{language}* |
| API Usage Description to Code Retrieval | the generated API or library usage description in *{language}* |
| Bug Description to Code Retrieval | 1. the modified code with one or more bugs
2. the generated bug description in *{language}* |
| Pseudocode to Code Retrieval | the generated pseudocode in *{language}* |
| Programming Tutorial Query to Code Example Retrieval | the generated programming tutorial query in *{language}* |
| Algorithm Description to Code Retrieval | the generated algorithm description in *{language}* |
| ***Code2Text Retrieval (10)*** | |
| Code Summary Retrieval | the generated summary in *{language}* |
| Code Review Retrieval | the generated review in *{language}* |
| Code Intent Retrieval | the generated intent in *{language}* |
| Code Optimization Retrieval | the generated optimization suggestions or performance analysis reports in *{language}* |
| Tutorial Retrieval | the generated tutorial in *{language}* |
| Code Issue Discussion Retrieval | 1. the generated buggy code
2. the generated error explanation in *{language}* |
| API Reference Retrieval | the generated API reference documentation in *{language}* |
| Code Walkthrough Retrieval | the generated walkthrough in *{language}* |
| Code Error Explanation Retrieval | the generated error explanation in *{language}* |
| Code to Requirement Retrieval | the generated requirement in *{language}* |
| ***Code2Code Retrieval (18)*** | |
| Code Context Retrieval | the generated piece of *{code_language}* code |
| Similar Code Retrieval | the generated piece of *{code_language}* code |
| Code Translation Retrieval | the generated piece of *{tgt_code_language}* code |
| Code Refinement Retrieval | the generated piece of *{code_language}* code |
| Secure Code Retrieval | the generated piece of *{code_language}* code |
| Code Version Update Retrieval | 1. the generated piece of *{code_language}* code
2. the generated piece of *{code_language}* code |
| Code Example Retrieval | the generated piece of *{code_language}* code |
| Code Dependency Retrieval | the generated piece of *{code_language}* code |
| Code Pattern Retrieval | the generated piece of *{code_language}* code |
| Code History Retrieval | the generated piece of *{code_language}* code |
| Code Integration Retrieval | the generated piece of *{code_language}* code |
| Optimized Code Retrieval | the generated piece of *{code_language}* code |
| Code Simplification Retrieval | the generated piece of *{code_language}* code |
| Code Modularization Retrieval | the generated piece of *{code_language}* code |
| Code Augmentation Retrieval | the generated piece of *{code_language}* code |
| Error Handling Retrieval | the generated piece of *{code_language}* code |
| Code Documentation Retrieval | the generated piece of *{code_language}* code |
| Library Adaptation Retrieval | the generated piece of *{code_language}* code |
| ***Hybrid Retrieval (9)*** | |
| Code Modification Retrieval | 1. the generated differences in *{language}* between the input code and output code
2. the generated modification instruction in *{language}* |
| Code Bug Fix Example Retrieval | 1. the generated buggy version of the code and a description in *{language}* of the bug or error
2. the generated piece of *{code_language}* code |
| Code Refactoring Pattern Retrieval | 1. the generated description of the desired refactoring goals or patterns in *{language}*
2. the generated piece of *{code_language}* code |
| Code Style Guideline Example Retrieval | 1. the generated query describing a desired coding style or best practice to improve it in *{language}*
2. the generated piece of *{code_language}* code |
| Code Migration Retrieval | 1. the generated specific migration requirement in *{language}*
2. the generated piece of *{code_language}* code |
| Code Optimization Hybrid Retrieval | 1. the generated question in *{language}* that requests a specific optimization for the code
2. the generated piece of *{code_language}* code |
| Code Comparison Retrieval | 1. the generated question in *{language}* about their differences or similarities
2. the generated response in *{language}* |
| Code Best Practices Retrieval | 1. the generated question in *{language}* about coding best practices related to the code
2. the generated response in *{language}* |
| Security Vulnerability Fix Retrieval | 1. the generated text description in *{language}* of a possible security concern in the code
2. the generated piece of *{code_language}* code |

Table 14: Annotation instructions for all 47 code retrieval tasks in CodeR-Pile, used in the annotation prompt.

| Task Name | Annotation Instruction |
|---|---|
| *Text2Code Retrieval (10)* | |
| Web Query to Code Retrieval | judge whether the code can help answer the web search query |
| Code Contest Retrieval | judge whether the code can help solve the code contest problem |
| Text to SQL Retrieval | judge whether the code is an appropriate response to the text query |
| Error Message to Code Retrieval | judge whether the code can help resolve the error message |
| Code Explanation to Implementation Retrieval | judge whether the code implements the functionality described in the explanation |
| API Usage Description to Code Retrieval | judge whether the code demonstrates the usage description of the API or library |
| Bug Description to Code Retrieval | judge whether the code can help address the described bug |
| Pseudocode to Code Retrieval | judge whether the code implements the procedure described in the pseudocode |
| Programming Tutorial Query to Code Example Retrieval | judge whether the code can answer the programming tutorial query |
| Algorithm Description to Code Retrieval | judge whether the code implements the algorithm described in the text |
| *Code2Text Retrieval (10)* | |
| Code Summary Retrieval | judge whether the text summarizes the code |
| Code Review Retrieval | judge whether the review explains the role of the code |
| Code Intent Retrieval | judge whether the text describes the intent of the code |
| Code Optimization Retrieval | judge whether the text provides optimization suggestions or performance analysis reports for the code |
| Tutorial Retrieval | judge whether the text is a tutorial or how-to guide that demonstrates how to use or implement similar code |
| Code Issue Discussion Retrieval | judge whether the text is a discussion or issue report related to the code |
| API Reference Retrieval | judge whether the text is an API reference documentation for the APIs or libraries used in the code |
| Code Walkthrough Retrieval | judge whether the text is a step-by-step walkthrough or detailed explanation of the code's logic and execution flow |
| Code Error Explanation Retrieval | judge whether the text describes potential errors or exceptions that may arise from the code |
| Code to Requirement Retrieval | judge whether the text is a software requirement or user story that the code fulfills |
| *Code2Code Retrieval (18)* | |
| Code Context Retrieval | judge whether the output code is the latter part of the input code |
| Similar Code Retrieval | judge whether the output code is semantically equivalent to the input code |
| Code Translation Retrieval | judge whether the output code is semantically equivalent to the input code |
| Code Refinement Retrieval | judge whether the output code is a refined version of the input code |
| Secure Code Retrieval | judge whether the output code is the version with enhanced security measures or vulnerability fixes compared to the input code |
| Code Version Update Retrieval | judge whether the output code is the version updated to comply with the syntax or features of a newer language version compared to the input code |
| Code Example Retrieval | judge whether the output code is the example code snippets that demonstrate how to use the library or API in the input code |
| Code Dependency Retrieval | judge whether the output code is the code segments that the input code depends on, including libraries, functions, and variables. |
| Code Pattern Retrieval | judge whether the output code follows the same design pattern or structure as the input code |
| Code History Retrieval | judge whether the output code is the historical version or iteration of the input code, and can help understand its development history. |
| Code Integration Retrieval | judge whether the output code demonstrates how to integrate the input code with other systems or components. |
| Optimized Code Retrieval | judge whether the output code is an optimized version of the input code |
| Code Simplification Retrieval | judge whether the output code is a simplified version of the input code |
| Code Modularization Retrieval | judge whether the output code is a modularized version of the input code |
| Code Augmentation Retrieval | judge whether the output code implements additional functionality while preserving the original behavior of the input code |
| Error Handling Retrieval | judge whether the output code incorporates error-checking or exception-handling mechanisms relevant to the input code |
| Code Documentation Retrieval | judge whether the output code contains inline comments or documentation explaining the functionality of the input code |
| Library Adaptation Retrieval | judge whether the output code achieves the same functionality using a different library or framework as the input code |
| *Hybrid Retrieval (9)* | |
| Code Modification Retrieval | judge whether the output code implements the requested modification described in the query |
| Code Bug Fix Example Retrieval | judge whether the output code fixes the bug or error described in the query. |
| Code Refactoring Pattern Retrieval | judge whether the output code exemplifies similar refactoring techniques or patterns described in the query |
| Code Style Guideline Example Retrieval | judge whether the output code adheres to the specified style guidelines or best practices described in the query |
| Code Migration Retrieval | judge whether the output code meets the migration requirement described in the query |
| Code Optimization Hybrid Retrieval | judge whether the output code implements the requested optimization described in the query |
| Code Comparison Retrieval | judge whether the response can answer the question described in the query |
| Code Best Practices Retrieval | judge whether the response can answer the question described in the query |
| Security Vulnerability Fix Retrieval | judge whether the output code addresses the security vulnerability described in the query |

Table 15: Query types and document types for all 47 code retrieval tasks in CodeR-Pile, used in the annotation prompt.

| Task Name | Query Type | Doc Type |
|---|---|---|
| *Text2Code Retrieval (10)* | | |
| Web Query to Code Retrieval | the web search query | the code |
| Code Contest Retrieval | the code contest problem | the code |
| Text to SQL Retrieval | the text query | the code |
| Error Message to Code Retrieval | the error message | the code |
| Code Explanation to Implementation Retrieval | the explanation | the code |
| API Usage Description to Code Retrieval | the API or library usage description | the code |
| Bug Description to Code Retrieval | the bug description | the code |
| Pseudocode to Code Retrieval | the pseudocode | the code |
| Programming Tutorial Query to Code Example Retrieval | the programming tutorial query | the code |
| Algorithm Description to Code Retrieval | the algorithm description | the code |
| *Code2Text Retrieval (10)* | | |
| Code Summary Retrieval | the code | the text |
| Code Review Retrieval | the code | the review |
| Code Intent Retrieval | the code | the text |
| Code Optimization Retrieval | the code | the text |
| Tutorial Retrieval | the code | the text |
| Code Issue Discussion Retrieval | the code | the text |
| API Reference Retrieval | the code | the text |
| Code Walkthrough Retrieval | the code | the text |
| Code Error Explanation Retrieval | the code | the text |
| Code to Requirement Retrieval | the code | the text |
| *Code2Code Retrieval (18)* | | |
| Code Context Retrieval | the input code | the output code |
| Similar Code Retrieval | the input code | the output code |
| Code Translation Retrieval | the input code | the output code |
| Code Refinement Retrieval | the input code | the output code |
| Secure Code Retrieval | the input code | the output code |
| Code Version Update Retrieval | the input code | the output code |
| Code Example Retrieval | the input code | the output code |
| Code Dependency Retrieval | the input code | the output code |
| Code Pattern Retrieval | the input code | the output code |
| Code History Retrieval | the input code | the output code |
| Code Integration Retrieval | the input code | the output code |
| Optimized Code Retrieval | the input code | the output code |
| Code Simplification Retrieval | the input code | the output code |
| Code Modularization Retrieval | the input code | the output code |
| Code Augmentation Retrieval | the input code | the output code |
| Error Handling Retrieval | the input code | the output code |
| Code Documentation Retrieval | the input code | the output code |
| Library Adaptation Retrieval | the input code | the output code |
| *Hybrid Retrieval (9)* | | |
| Code Modification Retrieval | the query | the output code |
| Code Bug Fix Example Retrieval | the query | the output code |
| Code Refactoring Pattern Retrieval | the query | the output code |
| Code Style Guideline Example Retrieval | the query | the output code |
| Code Migration Retrieval | the query | the output code |
| Code Optimization Hybrid Retrieval | the query | the output code |
| Code Comparison Retrieval | the query | the response |
| Code Best Practices Retrieval | the query | the response |
| Security Vulnerability Fix Retrieval | the query | the output code |

# F   Details on Data Process Pipeline

The prompt template for filtering query-positive pairs in the training data process phase is presented in Table 16.

Table 16: Prompt template for GPT filtering query-positive pairs. "*{Task Instruction}*" ∈ Table 7.

---

You are provided with a task, a query, and a document. Based on the task, determine whether the document can respond to the query.

Your response should be one of the following, along with an explanation:
- "Yes, simple" - If the document contains sufficient information to fully and directly respond to the query in a straightforward manner.
- "Yes, medium" - If the document contains sufficient information to fully and directly respond to the query, but the explanation or reasoning requires moderate effort.
- "Yes, hard" - If the document contains sufficient information to fully and directly respond to the query, but the explanation or reasoning requires significant effort or complexity.
- "No" - If the document does not contain any relevant information to respond to the query.

**Task:**
*{Task Instruction}*

**Query:**
*{query}*

**Document:**
*{document}*

Your output:

---

# G   Evaluation Instructions

Table 17 presents the instructions utilized in our evaluation of each task.

Table 17: Instructions for the evaluation process on the CoIR and CodeRAG benchmarks.

| Task Name | Instruction Template |
|---|---|
| ***CoIR*** | |
| Apps | Given a code contest problem description, retrieve relevant code that can help solve the problem. |
| CosQA | Given a web search query, retrieve relevant code that can help answer the query. |
| Text2SQL | Given a question in text, retrieve SQL queries that are appropriate responses to the question. |
| CSN | Given a piece of code, retrieve the document string that summarizes the code. |
| CSN-CCR | Given a piece of code segment, retrieve the code segment that is the latter part of the code. |
| CodeTrans-DL | Given a piece of code, retrieve code that is semantically equivalent to the input code. |
| CodeTrans-Contest | Given a piece of Python code, retrieve C++ code that is semantically equivalent to the input code. |
| StackOverFlow-QA | Given a question that consists of a mix of text and code snippets, retrieve relevant answers that also consist of a mix of text and code snippets, and can help answer the question. |
| CodeFeedBack-ST | Given a question that consists of a mix of text and code snippets, retrieve relevant answers that also consist of a mix of text and code snippets, and can help answer the question. |
| CodeFeedBack-MT | Given a multi-turn conversation history that consists of a mix of text and code snippets, retrieve relevant answers that also consist of a mix of text and code snippets, and can help answer the question. |
| ***CodeRAG*** | |
| HummanEval | Given a question that consists of a mix of text and code snippets, retrieve relevant answers that also consist of a mix of text and code snippets, and can help answer the question. |
| MBPP | Given a textual explanation of code functionality, retrieve the corresponding code implementation. |
| DS-1000 | Given a question that consists of a mix of text and code snippets, retrieve relevant answers that also consist of a mix of text and code snippets, and can help answer the question. |
| ODEX | Given a question, retrieve relevant answers that also consist of a mix of text and code snippets, and can help answer the question. |
| RepoEval | Given a piece of code segment, retrieve the code segment that is the latter part of the code. |
| SWE-bench-Lite | Given a code snippet containing a bug and a natural language description of the bug or error, retrieve code snippets that demonstrate solutions or fixes for similar bugs or errors (the desired documents). |

# H   Details for the Analysis of Synthetic Data

Table 18: Retrieval performance on the CoIR benchmark (NDCG@10) under different task coverage (RQ1), LLMs for generation (RQ2), and hard negatives strategy (RQ3).

| Model | Apps | CosQA | Text2SQL | CSN | CSN-CCR | CodeTrans | | StackOverFlow QA | CodeFeedBack | | Avg |
| --- | --- | --- | --- | --- | --- | --- | --- | --- | --- | --- | --- |
| | | | | | | -Contest | -DL | | -ST | -MT | |
| *Task Coverage* | | | | | | | | | | | |
| all data in 4 major task types | 53.18 | 32.39 | 47.38 | 73.76 | 92.96 | 87.43 | 35.36 | 92.73 | 81.25 | 80.00 | 67.64 |
| only Text-to-Code Retrieval data | 49.72 | 35.62 | 57.93 | 64.74 | 80.21 | 89.78 | 34.45 | 88.20 | 80.96 | 74.92 | 65.65 |
| *LLMs for Generation, using only 7 tasks* | | | | | | | | | | | |
| Qwen2.5-Coder-32B-Instruct | 44.55 | 28.52 | 60.01 | 70.19 | 91.51 | 87.80 | 32.83 | 90.65 | 80.98 | 85.86 | 67.29 |
| GPT-4o mini | 50.01 | 32.78 | 61.03 | 72.37 | 92.48 | 89.99 | 35.63 | 90.95 | 82.22 | 82.02 | 68.95 |
| *Hard Negatives Generation Strategy, using only 7 tasks* | | | | | | | | | | | |
| mine hard negatives | 44.55 | 28.52 | 60.01 | 70.19 | 91.51 | 87.80 | 32.83 | 90.65 | 80.98 | 85.86 | 67.29 |
| generate 7 hard negatives | 17.46 | 21.33 | 43.46 | 68.18 | 91.84 | 84.61 | 35.63 | 81.97 | 49.52 | 70.02 | 56.40 |
| generate 1 hard negatives | 21.25 | 25.13 | 44.57 | 70.50 | 90.06 | 88.62 | 34.90 | 86.91 | 59.17 | 72.66 | 59.38 |

