# OpenReview forum: "Towards A Generalist Code Embedding Model Based On Massive Data Synthesis"
_NeurIPS.cc/2025/Datasets_and_Benchmarks_Track — NeurIPS 2025 Datasets and Benchmarks Track poster_

### Official Review · Reviewer_G2xq · 2025-06-03

**Rating:** 5
**Confidence:** 4

**Summary:**

This paper introduces CodeR, a state-of-the-art general-purpose code embedding model designed for retrieval-augmented generation in software development. Its performance is enabled by CodeR-Pile, a large-scale synthetic dataset constructed under the DRU principle (Diversity, Reliability, Usability) through a multi-stage LLM-based pipeline. To effectively utilize heterogeneous training sources (text-only, mixed, and code-only data), the authors propose Annealing, a three-stage curriculum learning strategy that facilitates progressive knowledge transfer.

**Dataset Code Accessibility:**

Yes

**Ethical Considerations:**

No, there are no or only very minor ethics concerns

**Limitations Weaknesses:**

- CodeR is trained on a 1.5B parameter model, and while its performance is strong, the paper does not investigate scalability to larger model sizes. This leaves unanswered whether the observed gains would hold or improve further at larger scales (e.g., 7B or 13B), especially given comparisons with larger baselines like E5-Mistral-7B or CodeXEmbed-7B (Table 2). Exploring scaling behavior could enhance the model’s competitiveness in real-world high-performance RAG systems.
- While positive triplets achieve high annotation accuracy (93%), negative sample annotation accuracy is significantly lower (38%) for Qwen2.5-Coder (Table 6). This could lead to label noise and suboptimal contrastive training.
- Although the data pipeline aims for generalization, the use of task-specific instructions and hard-negative mining tailored to synthetic tasks (Sec. 4) could lead to implicit overfitting to artificial retrieval formats.

**Strengths Contributions:**

- The paper introduces CodeR-Pile, a large-scale synthetic dataset constructed using a principled, multi-stage LLM-driven pipeline.
- The authors present a three-stage data synthesis workflow involving large LLMs (e.g., GPT-4o, Claude 3) for task design and instruction generation, and lightweight models (e.g., Qwen2.5) for cost-effective triplet generation and annotation (Sec. 3).

---

> ### Author Rebuttal · Authors · 2025-07-30
>
> Dear Reviewer G2xq,
>
> Thank you very much for your thorough review and constructive feedback! In our response, we have carefully addressed your comments with the following discussions.
>
> > **W1:** *CodeR is trained on a 1.5B parameter model, and while its performance is strong, the paper does not investigate scalability to larger model sizes. This leaves unanswered whether the observed gains would hold or improve further at larger scales (e.g., 7B or 13B), especially given comparisons with larger baselines like E5-Mistral-7B or CodeXEmbed-7B (Table 2). Exploring scaling behavior could enhance the model’s competitiveness in real-world high-performance RAG systems.*
>
> - Thanks for your suggestion. We conducted an additional 1,000 training steps separately for both the Qwen2.5-Coder-1.5B and Qwen2.5-Coder-7B models. **The results on CoIR (****as** **presented in Table 1** **below****) confirm that larger models achieve better performance.** However, due to practical cost constraints, our current focus remains on training the 1.5B-parameter model. We plan to further explore larger models (7B and 14B) and release these versions in the future.
>
> |                    | 500step | 1000step |
> | ------------------ | ------- | -------- |
> | Qwen2.5-Coder-1.5B | 74.19   | 75.40    |
> | Qwen2.5-Coder-7B   | 75.88   | 77.03    |
>
> Table 1. The results on CoIR with different model sizes.
>
> > **W2:** *While positive triplets achieve high annotation accuracy (93%), negative sample annotation accuracy is significantly lower (38%) for Qwen2.5-Coder (Table 6). This could lead to label noise and suboptimal contrastive training.*
>
> - We adopt a relatively **stringent standard** for data filtering, which serves two purposes:
>   - **Ensuring data quality**: We retain only data that is both relevant and functionally meaningful, thereby ensuring a high-quality training set.
>   - **Prioritizing quality over quantity**: Although some correct data is inevitably filtered out, lowering the standard would allow low-quality data into the final dataset, while raising it further would reduce generation efficiency. By accepting this trade-off, we maintain a highly usable dataset.
> - Since the samples annotated as negative (label=0) are removed from the final CodeR-Pile dataset, they **do not contribute to label noise or lead to suboptimal contrastive training**.
> - Among the samples annotated as positive (label=1), 93% were confirmed to be correct under the stringent standard, demonstrating the high quality of the CodeR-Pile dataset and the effectiveness of our approach.
> - In conclusion, the overall quality of CodeR-Pile is very high, and we believe it can serve as a valuable resource for future research.
>
> > **W3:** *Although the data pipeline aims for generalization, the use of task-specific instructions and hard-negative mining tailored to synthetic tasks (Sec. 4) could lead to implicit overfitting to artificial retrieval formats.*
>
> - **The use of synthetic data does not compromise the model’s generalization ability; on the contrary, it enhances it**. When trained on synthetic data, the model consistently demonstrates strong performance across the CoIR and CodeRAG benchmarks, outperforming most baseline methods—even though it was not explicitly trained on these tasks. Furthermore, incorporating existing code retrieval data further improves performance, highlighting the valuable role of our synthetic data in boosting the model’s overall generalization capability.

---

### Official Review · Reviewer_aj4W · 2025-07-02

**Rating:** 4
**Confidence:** 3

**Summary:**

This work proposes a synthetic dataset using a curation pipeline called CODER-Pile, along with a code embedding model named CodeR, which utilizes the annealing technique. Evaluation against state-of-the-art baselines such as CoIR and CodeRAG shows significant improvement over various tasks, such as code retrieval.

**Dataset Code Accessibility:**

Yes

**Ethical Considerations:**

No, there are no or only very minor ethics concerns

**Final Justification:**

Thank you for your response. You have addressed the concerns I raised in my review; therefore, I will raise the score.

**Limitations Weaknesses:**

1. Based on the workflow illustrated in Figure 2, it appears that powerful large language models (LLMs) are utilized during the brainstorming and instruction phases, while lighter-weight LLMs are deployed in the generation pipeline. Could the authors please clarify their rationale behind selecting different LLMs for these stages? Specifically, what trade-offs or considerations guided the choice of heavier models for initial stages versus lighter models for generation?

2. Some important related works and strong baselines are missing from the evaluation, notably GraphCodeBERT[1] and CodeRL[2]. GraphCodeBERT serves as a foundational model for a wide range of code embedding tasks, making it a highly relevant and appropriate baseline for comparison. CodeRL introduces a generation pipeline that integrates dataset synthesis with code retrieval, representing a state-of-the-art approach in this area. However, neither of these papers is referenced, nor is there any experimental comparison against their methods, which limits the completeness of the evaluation.


[1] **[GraphCodeBERT (ICLR 2021)](https://arxiv.org/abs/2009.08366)**

[2] **[CodeRL (NeurIPS 2022)](https://arxiv.org/abs/2207.10397)**

**Strengths Contributions:**

1. The paper addresses an underexplored yet important area of research. While text retrieval has been widely studied, code retrieval remains relatively overlooked within the information retrieval community. This work helps fill that gap.

2. The evaluation is thorough and demonstrates solid improvements over strong baselines. The model shows clear gains in both standard code retrieval tasks and more challenging out-of-domain scenarios, highlighting its robustness.

3. The paper is well-written and easy to follow. The presentation is clear, with well-explained methods and results.

4. The choice of baselines is appropriate and competitive. The authors compare against strong benchmarks such as CoIR and CodeRAG, which helps validate the contributions of their approach.

---

> ### Author Rebuttal · Authors · 2025-07-30
>
> Dear Reviewer aj4W,
>
> Thank you very much for your thorough review and constructive feedback! In our response, we have carefully addressed your comments with the following discussions.
>
> > **W1:** *Based on the workflow illustrated in Figure 2, it appears that powerful large language models (LLMs) are utilized during the brainstorming and instruction phases, while lighter-weight LLMs are deployed in the generation pipeline. Could the authors please clarify their rationale behind selecting different LLMs for these stages? Specifically, what trade-offs or considerations guided the choice of heavier models for initial stages versus lighter models for generation?*
>
> - **The rationale behind using different LLMs at various stages is to balance data quality and production efficiency**.
>   - During the brainstorming and instruction phases, we use powerful LLMs because these steps establish the foundation for the entire dataset. Ensuring high-quality and meaningful task instructions is critical, so we apply strict control here, including powerful LLMs generation and rigorous verification by postgraduate computer science students.
>   - In contrast, the generation phase requires the production of a large volume of training data. Relying exclusively on powerful LLMs at this stage would be computationally expensive and limit scalability. Therefore, we utilize lighter-weight LLMs to efficiently generate data at scale.
> - **Our experiments further validate the effectiveness of this strategy**: Section 5.3 shows that the final synthetic data maintains high reliability, while Section 5.2 demonstrates that training on this data significantly improves experimental results.
>
> > **W2:** *Some important related works and strong baselines are missing from the evaluation, notably GraphCodeBERT[1] and CodeRL[2]. GraphCodeBERT serves as a foundational model for a wide range of code embedding tasks, making it a highly relevant and appropriate baseline for comparison. CodeRL introduces a generation pipeline that integrates dataset synthesis with code retrieval, representing a state-of-the-art approach in this area. However, neither of these papers is referenced, nor is there any experimental comparison against their methods, which limits the completeness of the evaluation.*
>
> Thanks for your valuable feedback. We appreciate the suggestion to include GraphCodeBERT and CodeRL in the revised version.
>
> - **GraphCodeBERT** is indeed a foundational model for code retrieval tasks. However, it is primarily designed for six specific programming languages and processes natural language and code separately. In contrast, our evaluation includes a broader range of programming languages and introduces datasets where natural language and code are mixed within the same input. Since GraphCodeBERT does not directly support such a complex scenario, it is not fully comparable as a baseline in our setting. Nevertheless, we acknowledge its importance and will include a discussion of GraphCodeBERT in the revised version.
> - **CodeRL** focuses on code generation by combining pre-trained models with reinforcement learning to optimize output quality. This approach is effective for code generation tasks, but since our work targets a different problem — code retrieval — its methodology is not directly comparable. We will include a description of CodeRL in the related work section to provide readers with a broader context.
>
> In summary, we will add the following content to the revised version:
>
> > In code generation, methods like CodeRL integrate reinforcement learning with pre-trained models to enhance the generation of executable code. Similarly, CodeT generates multiple code samples and selects the best solution using test cases to boost performance. In contrast, research on code retrieval is less mature. Models such as CodeBERT and GraphCodeBERT are limited by the number of supported languages and their ability to handle complex retrieval scenarios. Therefore, there is a pressing need to develop more robust and versatile code retrieval models.

---

### Official Review · Reviewer_yUyZ · 2025-07-07

**Rating:** 5
**Confidence:** 3

**Summary:**

This paper introduces CodeR-Pile, a large-scale, synthetic dataset designed to train general-purpose code embedding models. The authors identify the scarcity of high-quality data as a key limitation for current code retrieval models and propose a novel, three-stage data synthesis pipeline guided by a "DRU" (Diversity, Reliability, Usability) principle to address this gap. The resulting dataset contains 2.9 million samples spanning 47 distinct code-related tasks, 20 programming languages, and two natural languages. To demonstrate the dataset's utility, the authors use it to train a new embedding model, CodeR, which they report achieves state-of-the-art performance on several code retrieval benchmarks.

**Additional Feedback:**

The creation of a large-scale dataset for code retrieval is a valuable endeavor. However, the major concern regarding the flawed quality control process must be addressed. A dataset's value is predicated on its reliability, and a 38% accuracy for filtering out incorrect samples is not acceptable.

I would recommend the authors focus on a more rigorous and verifiable quality control mechanism, perhaps using multiple models for consensus or incorporating more human-in-the-loop validation. Simplifying the overall pipeline would also improve the work's clarity and reproducibility.

**Dataset Code Accessibility:**

Partly

**Dataset Code Comments:**

While the dataset is hosted on a public platform (Hugging Face), its accessibility and integrity appear to have significant issues. First, the automated compliance report provided for this submission notes a major accessibility problem: 50 data files were inaccessible at the time of review. For a dataset submission, this level of incompleteness is a serious concern.

Furthermore, the dataset's hosting page itself issues a safety warning, indicating that "This dataset has 7 files scanned as unsafe". The combination of numerous inaccessible files and additional files flagged as unsafe raises serious questions about the dataset's readiness for release and its overall reliability. These issues hinder independent verification and must be rectified to ensure the dataset is fully and safely available to the community.

**Ethical Comments:**

The work is based on generating synthetic data from publicly available code on GitHub. The dataset is released under a non-commercial license, which is appropriate for a research artifact. There are no obvious ethical concerns regarding data privacy or malicious use.

**Ethical Considerations:**

No, there are no or only very minor ethics concerns

**Final Justification:**

The reviewers provide justification for each step of the data creation process, and they all make sense to me. After a second look, I found that even though some of the steps are not formal enough, they are still valuable for the research community and can provide a solid benchmark for the code embedding models. Based on that, I raised my score to accept.

**Limitations Weaknesses:**

Despite the impressive scale, I have significant concerns regarding the dataset's creation process and the reliability of its contents.

1. Fundamentally Flawed Reliability Check: The "Reliability" claim of the DRU principle is severely undermined by the paper's own analysis. Table 6 shows that the lightweight LLM used for quality control is only 38% accurate at identifying false positives (annotating pairs with label=0).  This is a critical flaw, as it implies a substantial amount of noisy, incorrect data has likely been incorporated into the final CodeR-Pile dataset, calling the overall quality of the resource into question.

2. Overly Complex and Opaque Curation Process: The complete data generation and processing workflow (Figures 2 and 3) is exceedingly complex.  It involves multiple LLMs, multi-stage filtering, and a bespoke curriculum learning strategy ("Annealing") that is difficult to untangle. This over-engineering makes it hard to understand the true properties of the final dataset and makes the curation process itself difficult for others to faithfully replicate or adapt.

3. Unclear Value of a Purely Synthetic Dataset: While the authors demonstrate that a model trained on this data performs well on other benchmarks, the intrinsic value and characteristics of a purely synthetic dataset of this nature remain underexplored. The paper lacks a sufficient discussion of how well the synthetic data distribution of CodeR-Pile mirrors real-world code retrieval scenarios and does not analyze the potential biases or artifacts introduced by the LLM-based generation process.

**Strengths Contributions:**

The primary contribution of this work is the CodeR-Pile dataset itself, which is a significant artifact for the code intelligence community. I find its key strengths to be:

1. Scale and Diversity: CodeR-Pile is impressive in its scale and diversity. With 2.9 million samples covering 47 tasks and 20 programming languages (as detailed in Table 1 and Figure 1), it is one of the most comprehensive public resources for code retrieval research to date.  This breadth is a valuable asset for training more generalist models.

2. Systematic Data Synthesis Pipeline: The paper proposes a structured, three-stage pipeline for data synthesis (Figure 2).  The conceptual framework of "Brainstorming," "Instructing," and "Triplets Generation" is a systematic approach to creating synthetic data at scale. The design choice to use powerful LLMs for creative, high-level planning and more cost-effective lightweight LLMs for bulk generation is a clever and practical methodology.

3. Public Availability and Documentation: A major strength is the public release of the dataset on a platform like Hugging Face, along with extensive documentation in the appendices. The detailed tables outlining every task, instruction, and prompt (e.g., Table 8, 13-16 in appendix) are commendable and crucial for enabling the community to understand, use, and build upon this work.

---

> ### Author Rebuttal · Authors · 2025-07-30
>
> Dear Reviewer yUuZ,
>
> Thank you very much for your thorough review and constructive feedback! In our response, we have carefully addressed your comments with the following discussions.
>
> > **W1:** *Fundamentally Flawed Reliability Check: The "Reliability" claim of the DRU principle is severely undermined by the paper's own analysis. Table 6 shows that the lightweight LLM used for quality control is only 38% accurate at identifying false positives (annotating pairs with label=0). This is a critical flaw, as it implies a substantial amount of noisy, incorrect data has likely been incorporated into the final CodeR-Pile dataset, calling the overall quality of the resource into question.*
> >
> > **F1:** *The creation of a large-scale dataset for code retrieval is a valuable endeavor. However, the major concern regarding the flawed quality control process must be addressed. A dataset's value is predicated on its reliability, and a 38% accuracy for filtering out incorrect samples is not acceptable.*
>
> - We adopt a relatively **stringent standard** for data filtering, which serves two purposes:
>   - **Ensuring data quality**: We retain only data that is both relevant and functionally meaningful, thereby ensuring a high-quality training set.
>   - **Prioritizing quality over quantity**: Although some correct data is inevitably filtered out, lowering the standard would allow low-quality data into the final dataset, while raising it further would reduce generation efficiency. By accepting this trade-off, we maintain a highly usable dataset.
> - Since the samples annotated as negative (label=0) are removed from the final CodeR-Pile dataset, they **do not contribute to label noise or introduce incorrect data**.
> - Among the samples annotated as positive (label=1), 93% were confirmed to be correct under the stringent standard, demonstrating the high quality of the CodeR-Pile dataset and the effectiveness of our approach.
> - In conclusion, the overall quality of CodeR-Pile is very high, and we believe it can serve as a valuable resource for future research.
>
> > **W2:** *Overly Complex and Opaque Curation Process: The complete data generation and processing workflow (Figures 2 and 3) is exceedingly complex. It involves multiple LLMs, multi-stage filtering, and a bespoke curriculum learning strategy ("Annealing") that is difficult to untangle. This over-engineering makes it hard to understand the true properties of the final dataset and makes the curation process itself difficult for others to faithfully replicate or adapt.*
>
> - **The data generation process** **is simple and straightforward, it** **only consists of four main steps:** 1) Sample a code file and clean it. 2) Prompt the LLM to generate a query-positive pair using the generation prompt. 3) Prompt the LLM to annotate the generated query-positive pair using the annotation prompt. 4) Mine hard negatives for the query-positive pairs annotated with a positive relevance label (label = 1).
> - In addition, there are two key operations in the full data generation process:
>   - **Brainstorming**: This operation aims to improve the **diversity** of the generated data by covering a wider range of code retrieval tasks.
>   - **Instructing**: This operation aims to improve the **quality** of the generated data by ensuring that high-quality and meaningful task instructions are used in the generation process.
> - **The curriculum learning strategy (Annealing) is simple and straightforward too, which only consists of three main steps**: 1) Train on text-only data. 2) Train on text and code data. 3) Train on challenging code data. The process of obtaining challenging code data is straightforward, as it only requires filtering the code data. This strategy is specifically designed for model fine-tuning, rather than serving as a general data processing pipeline for code data.
> - Overall, our data generation process and curriculum learning strategy are both **simple and easy to follow**, making them easy for others to replicate and adapt.
>
> > **W3:** *Unclear Value of a Purely Synthetic Dataset: While the authors demonstrate that a model trained on this data performs well on other benchmarks, the intrinsic value and characteristics of a purely synthetic dataset of this nature remain underexplored. The paper lacks a sufficient discussion of how well the synthetic data distribution of CodeR-Pile mirrors real-world code retrieval scenarios and does not analyze the potential biases or artifacts introduced by the LLM-based generation process.*
>
> - **We guarantee that the generated data in CodeR-Pile is highly relevant to real-world code retrieval scenarios**. To ensure its real-world relevance and quality, we took the following steps:
>   - **Grounding in real-world data**: We generate new code retrieval tasks starting from *real-world seed tasks* and use a *clean github code corpus* as the basis, ensuring the data reflects realistic code patterns and scenarios.
>   - **Quality verification**: Our data generation pipeline undergoes both *automated LLM-based checks* and *human verification* by experienced postgraduate computer science students. We guarantee that the new code retrieval tasks are close to real-world scenarios and that the generation instructions used for LLMs are of high quality. The tasks are presented in Appendix D of our paper and the instructions are available in Appendix E of our paper.
>   - **Empirical Validation**: Table 1 below shows that models trained on synthetic data alone outperform strong baselines (70.7% vs. 64.6% under OOD setting), and the combination of synthetic and existing real code retrieval data further boosts performance from 70.9% to 72.8%. This demonstrates the synthetic data’s practical value and alignment with real-world tasks.
> - While we acknowledge potential biases from LLM generation, our verification pipeline and the use of real-world seed tasks substantially mitigate these biases.
>
> |                                                              | CodeRAG    |
> | ------------------------------------------------------------ | ---------- |
> | CodeXEmbed-2B (general)                                      | 64.6 (OOD) |
> | Ours (text-only data & synthetic data)                       | 70.7 (OOD) |
> | Ours (text-only data & existing code retrieval data)         | 70.9 (OOD) |
> | Ours (text-only data & existing code retrieval data & synthetic data) | 72.8 (OOD) |
>
> Table 1. Highlighted experiment results to demonstrate the value of CodeR-Pile dataset.
>
> > **C1:** *While the dataset is hosted on a public platform (Hugging Face), its accessibility and integrity appear to have significant issues. First, the automated compliance report provided for this submission notes a major accessibility problem: 50 data files were inaccessible at the time of review. For a dataset submission, this level of incompleteness is a serious concern.*
>
> - **In our network environment, all datasets are fully accessible**. You can use the script provided below to access the full data:
>
> ```Python
> from datasets import load_dataset_builder, load_dataset
>
> dataset_name = "nebula2025/CodeR-Pile"
> cache_dir = ".cache"
>
> configs = []
> try:
>     builder = load_dataset(dataset_name, cache_dir=cache_dir)
> except ValueError as e:
>     e = str(e)
>     configs = eval(e[e.index('['): e.index(']') + 1])
>
> for config in configs:
>     dataset = load_dataset(dataset_name, config, cache_dir=cache_dir, download_mode='force_redownload')
> ```
>
> > **C2:** *Furthermore, the dataset's hosting page itself issues a safety warning, indicating that "This dataset has 7 files scanned as unsafe". The combination of numerous inaccessible files and additional files flagged as unsafe raises serious questions about the dataset's readiness for release and its overall reliability. These issues hinder independent verification and must be rectified to ensure the dataset is fully and safely available to the community.*
>
> - **This "unsafe" warning is actually a false alarm from Huggingface, as noted previously within the community**. Because our corpus is fully inherited from `codeparrot/github-code-clean`, it also inherits these false "unsafe" flags. Here is the explain in community: "I did a little experiment with the Parquet files detected as being infected. Scanning each Parquet file with ClamAV as a single input file does show infections. However, if I break out all of the `code` column contents from one of those Parquet files into separate files, then scan the directory containing all of those files, no infections are detected."
>
> > **F2:** *I would recommend the authors focus on a more rigorous and verifiable quality control mechanism, perhaps using multiple models for consensus or incorporating more human-in-the-loop validation. Simplifying the overall pipeline would also improve the work's clarity and reproducibility.*
>
> - **We agree that a rigorous and verifiable quality control mechanism is essential**. Therefore, our current verification process includes two key aspects: model-based validation, and review and editing conducted by several postgraduate computer science students, which ensure high-quality data. We also plan to further enhance our human-in-the-loop validation in the future.
> - Regarding the overall pipeline, as mentioned above, it is designed to be simple and straightforward, making it easy to implement and reproduce.

---

> > ### Author Response · Authors · 2025-08-05
> >
> > Dear Reviewer yUuZ,
> >
> > Thank you for your thoughtful feedback. In our response, we have carefully addressed each of the concerns you raised. In particular:
> >
> > * The 38% false negative rate refers to the filtered-out data; the samples included in the final dataset are highly reliable.
> > * The data generation process is simple and consists of only four main steps.
> > * CodeR-Pile is closely aligned with real-world scenarios, as demonstrated by compelling results.
> > * All datasets are fully accessible via the script we have provided.
> > * The "unsafe" warning appears to be a false alarm.
> >
> > As the rebuttal stage is coming to its conclusion, we would greatly appreciate your feedback on whether there are any outstanding concerns. We remain committed to addressing all issues to the best of our ability.
> >
> > Thank you again for your time and consideration!
> >
> > Sincerely, \
> > The Authors

---

> > > ### Comment · Reviewer_yUyZ · 2025-08-05
> > >
> > > Thank you for your feedback and it addressed most of my concerns. Based on that, I raised the score accordingly. Good luck!

---

### Official Review · Reviewer_DVgM · 2025-07-09

**Rating:** 4
**Confidence:** 4

**Summary:**

The paper presents CodeR, an embedding model for general purpose code retrieval tasks. The paper also open-source CodeR-Pile, a large-scale synthetic dataset that is generated via a novel data synthesis pipeline. To train the embedding model, the authors propose a curriculum learning strategy that enables effective knowledge transfer across source of data. CodeR is evaluated on 16 diverse tasks and authors show that CodeR significantly outperforms existing baselines.

**Additional Feedback:**

- Consider comparing with CodeSage (https://code-representation-learning.github.io/codesage-v2.html) as they have a 1.3B model is of similar size of CodeR-1.5B.
- Consider converting some result tables into plots. Currently there are too many tables and too many numbers which are difficult to perceive, specially understanding the takeaway message. Also consider writing more verbose table captions which summarizes the main point (if possible).

**Dataset Code Accessibility:**

Yes

**Ethical Considerations:**

No, there are no or only very minor ethics concerns

**Final Justification:**

I am sticking to my score, primarily because I still have concerns regarding data contamination and keyword based matching may not capture the contamination issue.

**Limitations Weaknesses:**

- From Table 2, it seems performance improvements are not consistent across datasets. For example, in Apps dataset, the performance of CodeR looks very high compared to CodeXEmbed. This raises the concern of data contamination. Specially we see at the delta between "w/ text-only data & existing code retrieval data" and "CodeR-1.5B + text-only data & full code data" for APPS.

- Too many results tables/numbers that make it difficult to understand the value of critical components, presented in the analysis section. The paper needs to improve readability, specially in the experiment results section.

**Strengths Contributions:**

- CodeR-Pile, a synthetic dataset spanning 20 languages, covering 47 tasks, including 2.9M samples, should significantly benefit future works to train LLMs to get better embedding model. Overall, CodeR-Pile seems to be generated with a thoughtfully designed pipeline that includes three stages of brainstorming, instructing, and generation of triplets.
- The paper presents a curriculum learning strategy which they referred as "Annealing" also shown to be effective for training.
- The experiment results show significant improvements over the baseline. The ablation and analysis addresses the impact of task coverage in the CodeR-Pile dataset, LLMs as data generator, impact of mined hard negatives vs. LLM-generated hard negatives, and how reliably LLMs generate relevance data.
- The paper is overall well written.

---

> ### Author Rebuttal · Authors · 2025-07-30
>
> Dear Reviewer DVgM,
>
> Thank you very much for your thorough review and constructive feedback! In our response, we have carefully addressed your comments with the following discussions.
>
> > **W1:** *From Table 2, it seems performance improvements are not consistent across datasets. For example, in Apps dataset, the performance of CodeR looks very high compared to CodeXEmbed. This raises the concern of data contamination. Specially we see at the delta between "w/ text-only data & existing code retrieval data" and "CodeR-1.5B + text-only data & full code data" for APPS.*
>
> - First, **APPS is a relatively simple task which can be efficiently learned from the training data**. The APPS dataset in CoIR [1] benchmark has the following features: 1) each query corresponds to only one positive document; 2) the corpus only contains 8765 documents; 3) the positive documents are relatively independent from each other. The three features make it relatively easy to optimize embedding models' performance.
>
> - Second, **we carefully verify that there is no data contamination through a string-match-based query overlap analysis**. Specifically, for each test query q_t in APPS, we retrieve the Top-20 most relevant training queries in *full code data* using BM25, and then compute the weighted Jaccard similarity [2] between q_t and each retrieved query q_i. Only 9 queries have a similarity score above 0.60, and all originate from the APPS training set in CoIR. Their IDs and similarity scores are listed in Table 1. As these queries represent a small portion of the test set, their impact on performance is negligible.
>
> |Training Query ID in CoIR|Test Query ID in CoIR|Weighted Jaccard Similarity|
> |-|-|-|
> |q20|q6006|0.95|
> |q2281|q7129|0.61|
> |q68|q7234|0.61|
> |q48|q7556|0.66|
> |q1920|q7632|0.62|
> |q2130|q7930|0.94|
> |q2153|q8030|0.73|
> |q2390|q8687|0.90|
> |q2305|q8715|0.63|
>
> Table 1. Data contamination analysis results on APPS.
>
> [1] CoIR: A Comprehensive Benchmark for Code Information Retrieval Models.
>
> [2] Improved Consistent Sampling, Weighted Minhash and L1 Sketching.
>
> > **W2:** *Too many results tables/numbers that make it difficult to understand the value of critical components, presented in the analysis section. The paper needs to improve readability, specially in the experiment results section.*
> >
> > **F2:** *Consider converting some result tables into plots. Currently there are too many tables and too many numbers which are difficult to perceive, specially understanding the takeaway message. Also consider writing more verbose table captions which summarizes the main point (if possible).*
>
> - Thanks for your suggestion. We will consider using bar charts to present the results in the revised version, comparing our method with representative baselines to better highlight differences across different datasets. We will also summarize the main points in the table captions.
>
> > **F1:** *Consider comparing with CodeSage as they have a 1.3B model is of similar size of CodeR-1.5B.*
>
> - Thanks for your suggestion. As shown in Table 2, we have compared our model with CodeSage-large-v2 and found that our model outperforms CodeSage-large-v2 of the same scale. We will also include this comparison in the revised version.
>
> | |**Size**|**CoIR (Avg)**|**CodeRAG (Avg)**|
> |-|-|-|-|
> |CodeR-1.5B (Ours)|1.5B|81.77|72.8|
> |CodeSage-large-v2|1.3B|64.08|58.6|
>
> Table 2. Comparison between our model CodeR-1.5B and CodeSage-large-v2.

---

### Author Response · Authors · 2025-08-09

Dear Reviewers, AC, SAC, and PC,

We sincerely thank you for your careful reviews and insightful feedback. We are glad that all reviewers recognized the key strengths of our paper, including:

- The CodeR-Pile dataset is a large-scale, valuable synthetic dataset with a novel and effective data synthesis pipeline.
- The “Annealing” curriculum learning strategy is effective for training the code retrieval model.
- The evaluation is thorough and demonstrates solid improvements over strong baselines.

**We are particularly encouraged by the highly positive feedback from reviewers DVgM, aj4W, and G2xq from the beginning of the rebuttal phase. We have carefully addressed all reviewers’ concerns and questions, and we appreciate the willingness of reviewers DVgM and yUyZ to update their initial assessments.** The additional analyses and explanations provided in our rebuttal have further strengthened our work.

We thank all reviewers for their invaluable guidance. In the final version, we will incorporate the supplementary results discussed during the rebuttal.

Thank you again for your time and thoughtful comments.

---

### Decision · Program_Chairs · 2025-09-18

**Decision:**

Accept (poster)

**Comment:**

Thank you for your submission to NeurIPS 2025 Datasets and Benchmarks.
We are pleased to inform you that all reviewers have agreed to accept your paper.

The reviewers recognized that the paper presents a significant contribution to the code intelligence community by introducing CodeR-Pile, a large-scale, diverse synthetic dataset for code retrieval, and the CodeR embedding model.

However, please carefully address the reviewers' comments in the camera-ready version.